# An attempt to explain recent changes in European snowfall extremes

Davide Faranda[1,2,3]

[1]Laboratoire des Sciences du Climat et de l'Environnement, UMR 8212 CEA-CNRS-UVSQ, Université Paris-Saclay, IPSL, 91191 Gif-sur-Yvette, France
[2]London Mathematical Laboratory, 8 Margravine Gardens London, W6 8RH, UK
[3]LMD/IPSL, Ecole Normale Superieure, PSL research University, Paris, France

**Correspondence:** Davide Faranda (davide.faranda@lsce.ipsl.fr)

**Abstract.** The goal of this work is to investigate and explain recent changes in total and maximum yearly snowfall from daily data in light of the current global warming or the interdecadal variability of the atmospheric circulation. We focus on the period 1979-2018 and compare two different data-sets: the ERA5 reanalysis data and the E-OBSv20.0e data, where snowfall is identified from rainfall by applying a threshold on temperature. We compute changes as differences from quantities computed for the periods 1999-2018 and 1979-1998. On one hand, we show that the decline in average snowfall observed in almost all European regions is coherent with previous findings and can be linked to global warming. On the other hand, we observe contrasting changes in maxima and sometimes disagreement in the sign of changes in the two data-sets. Coherent positive trends are found for few countries in the Balkans. These have been investigated in details by looking at modifications in the atmospheric weather patterns as well as local thermodynamic factors concurring to large snowfall events. We link these changes to the stronger prevalence of Atlantic ridge or Blocking patterns associated with deeper cyclonic structure over the Adriatic or the Tyrrhenian sea. These cyclones find warmer surfaces and large availability of humidity and CAPE, thus producing large snowfall amounts, enhanced by Stau effect on the Balkans topography.

## 1 Introduction

Heavy snowfalls can have a great impact on economy and society. In January 2017, a cold spell affected most of Eastern and Central Europe and part of southern Europe, causing the death of at least 60 people: The combination of snowfalls with a series of earthquake in Central Italy caused a disastrous avalanche that hit the town of Rigopiano in Abruzzo where a landslide swept and destroyed a hotel, causing several casualties (Frigo et al., 2018). On January 8th, accumulations of 22-23 cm have been measured in some points on the beach of Porto Cesareo, in Apulia. Inland, snow reached and exceeded 2 meters in height on the Apennines. Two further recent examples of snowfalls affecting large populated areas are the February/March 2012 snowstorm in northern Italy with up to 50 cm of snowfall measured in Bologna (Bisci et al., 2012), and the winter 2018 snowstorm Emma, which affected UK with up to 40cm snowfall in Wales and the disruption of air and rail transportation in London, Manchester

and Liverpool areas (Tonks, 2018).

Besides their cost in terms of societal and economical impacts, these extreme events are often invoked by climate change denial groups to mystify the public opinion (Revkin, 2008) and it is therefore important to understand why, in an undeniable context of climate change, we do not observe a sharp decrease of their frequency/intensity. Indeed, although global temperature rise has driven an overall decrease of average snowfall in past decades (Déry and Brown, 2007) and this decreasing trend is expected to continue in future "business-as-usual" emission scenarios (Brutel-Vuilmet et al., 2013), it is not clear whether the
same conclusions hold for extreme snowfall events. Atmospheric extreme weather events do not always have a trivial relation with average global warming (Murray and Ebi, 2012). The goal of this paper is to shed a light on recent changes in the dynamics of extreme snowfalls, by projecting the recent changes in frequency/intensity of extreme snowfalls on the large scale (synoptic) dynamical drivers and identifying possible small scale convective thermodynamic feedback.

The focus of this study is to understand changes in daily heavy snowfalls at the scale of European regions and countries. Daily extreme snowfalls result from the interplay of both dynamical and thermodynamic factors, acting at different spatial and time scales: at local (few kms, few hours) scales, geographical features and convection may enhance snowfall precipitations. Persistence of convective snowfalls for several hours on the same region can provide large snowfall amounts detectable at daily time-scales. At synoptic scales, snowfalls are driven by extratropical cyclones ( ~1000 km, 2-6 days) traveling southwards
in jet-stream meanders formed by the disruption of the normal westerly flow (Tibaldi and Buzzi, 1983; Barnes et al., 2014; Lehmann and Coumou, 2015). Oscillations of the jet stream are associated with low-frequency variability of weather patterns that can modulate daily synoptic fields and snowfall events. (Wallace and Hobbs, 2006). These conditions create a dipole consisting of high pressure structures over some regions and low pressure systems (extratropical cyclones) travelling southward in other regions. The most common way to link low frequency variability to weather phenomena is the computation of daily
weather regimes (Vautard, 1990; Michelangeli et al., 1995). In Yiou and Nogaj (2004), first connections between extreme weather events and weather regimes have been established. Madonna et al. (2017) found a clear link between eddy-driven jet variability and weather regimes in the North Atlantic-European sector. In winter, if blocking high pressure becomes established close to Greenland, cold air from polar latitudes can be advected towards western Europe (North Atlantic Oscillation negative phase (Cattiaux et al., 2010)). When this weather pattern is associated with extratropical cyclones travelling southward from
northern latitudes extreme snowfalls over UK, France, Benelux and the Iberian Peninsula are expected. If a high pressure ridge (Atlantic Ridge) extends from the Azores Islands towards the Icelandic region or the British isles, cold air coming from Russia or Scandinavia flows in the Mediterranean Sea. This can cause cyclogenesis in the Tyrrhenian (Genoa lows) or in the Adriatic seas triggering extreme snowfalls over Italy, the Balkans, Greece and Turkey (Buehler et al., 2011).

When looking at long-term decadal trends in snowfalls, the analysis of weather patterns can provide important information to assess whether the changes in frequency and intensity are due to long term variability of the atmospheric circulation or induced by anthropogenic forcing (Strong et al., 2009; Overland and Wang, 2010; Woollings et al., 2010; Wu and Zhang,

2010; Deser et al., 2017). So far, it has been very difficult to prove any significant shift in the dynamical patterns observed at mid-latitudes (Shepherd, 2014). On one side, Cohen et al. (2014) and Kim et al. (2014) showed that the recent increase of temperatures in the Arctic is associated with an amplification of planetary waves, affecting storm tracks and leading to enhanced winter conditions. On the other hand, several authors found a zonalization of the mid-latitude flow (Lorenz and DeWeaver, 2007; Chen et al., 2016; Screen et al., 2014; Faranda et al., 2019) and a minimal or even undetectable effect of the Arctic sea-ice on the meandering of the jet at mid-latitudes (Blackport et al., 2019; Screen, 2017; Screen et al., 2018).

Although heavy snowfalls are driven by the large scale atmospheric circulation, their effects can be greatly enhanced by local geographic constraints and thermodynamic feedbacks (Lüthi et al., 2019; Bartolini, 2019). Local features like the Alps in Europe the Great Lakes in USA or the topography of Japanese islands may increase precipitation and provide relevant feedback to extreme snowfalls (Niziol et al., 1995). For Japan, Kawase et al. (2016) have shown that thermodynamic feedbacks from anthropogenic forcing may enhance extreme snowfalls in future climates via the interaction of the Japan Sea polar air mass convergence zone with the topography. A similar mechanisms exist also for the Mediterranean sea, as recently detailed in D'Errico et al. (2019). The mid-tropospheric cold winter air advection associated with the synoptic patterns flows over the relatively warmer waters of the Mediterranean sea picks up water vapor from the water surface. This warmer and wetter air rises and cools as it moves away from the sea towards land areas forming convective clouds that transform moisture into snow. In the mountainous topography of the European continent, this phenomenon can be extremely powerful in triggering heavy snowfalls (Beniston et al., 2018; Bartolini, 2019) even in future climate warming scenarios (D'Errico et al., 2019). We will also consider this effect in driving convection via the analysis of convective available potential energy patterns during extreme events.

The paper is organized as follows. In section 2, we describe the data-sets used in this study and the difficulties arising in assessing the quality of snow data. In section 3 we compute the changes in snowfall extremes and discuss their consistency among the data-sets. In section 4 we focus on those countries showing an increase of maximum snowfall and explain these changes in light of the thermodynamics and the dynamics of the atmosphere at daily scales. Conclusions and limitations of this study are presented in section 5.

## 2    Data and Methods

Good quality snow data at synoptic or regional scales are difficult to obtain (Rasmussen et al., 2012). From an observational point of view, quality observational data-sets exist only at high mountains sites and in regions where snowfalls are recurrent phenomena. Excellent snow data-sets exist for Scandinavian countries as well as for the Alpine regions (Auer et al., 2005; Scherrer and Appenzeller, 2006; Isotta et al., 2014). Our goal is however to study changes in snowfall at a European level, not limiting our analysis to mountain areas but also to those regions where these phenomena are rare. We have therefore to rely on reanalyses as well as on gridded observational data. In this study we analyse the period 1979-2018 and use a reanalysis product

(European Centre for Medium-Range Weather Forecast (ECMWF) Reanalysis 5th Generation product ERA5) as well as gridded observations data-set (European Climate Assessment Data E-OBSv20.0e). The reference data-set will be ERA5 (C3S), a very recent product by the ECMWF with high resolution (0.25° horizontal resolution) and accurate physical parametrizations. For the observations, we use E-OBSv20.0e (0.25 ° horizontal resolution) which contains gridded temperatures and precipitations observations (Cornes et al., 2018).

Another problem in comparing snow data issued from different sources is the choice of the variable associated with precipitating snow (Nitu and Wong, 2010). Snow precipitations can both be measured as snowfall (SF), or from snow-depth on the ground. Both the measurements have pros and cons. Snowfall is obtained by melting snow falling inside a heated rain gauge and it is expressed in $Kg/m^2$ or cm. An advantage of using this variable is the accuracy of the measurement. For obvious reasons, SF is mostly used by hydrologists as it has a direct connection with runoff and rivers discharge. Since the snow is immediately transformed into water, SF does not distinguish between snowfalls which produce accumulations on the ground or not. Snow depth is a measure of the snow height on the ground and it can be affected by several problems due to gravitational settling, wind packing, melting and re-crystallization. In this paper we will therefore use daily SF and express it in cm. We now explain how to get this quantity from the different data-sets considered in this study.

- For ERA5, we use the accumulated total snowfall that has fallen to the Earth's surface. This quantity consists of both snow due to the large-scale atmospheric flow and convective precipitations. It measures the total amount of water accumulated from the beginning of the forecast time to the end of the forecast step. The units given measure the depth the water would have if the snow melted and was spread evenly over the grid box. We get the snowfall from hourly data and construct the daily SF by summing up the snowfall in intervals of 24 hours. We chose ERA5 data-set as the preferential one for our study because of its physical consistency and the use of advanced assimilation techniques for its compilation.

- For E-OBSv20.0e [40.375W-50E,25.375N-75.375N] only land points, we do not dispose directly of snowfall data. We have to infer them from daily total precipitation and daily mean temperature data. We apply a simple algorithm which consists of considering as SF all precipitations occurred in days where the average temperature is below 2° C. Of course with this method we can have false positive as well as false negative events, but we have verified (not shown) that results do not depend qualitatively from the threshold providing that it is chosen between 0° C and 2.5° C. Since we use a threshold of 2° C, some of the precipitation would not be snowfall.

We now present the climatology for the two data-sets used in this study and focus on two quantities: yearly total snowfall SF (average 1979-2018) in Figure 1 and the maximum yearly snowfall SF from daily data (average 1979-2018) in Figure 2. We show results at two different levels, taken from the 2016 nomenclature of territorial units for statistics (NUTS) of the European union at regional (NUTS-2) and national (NUTS-0) level Commission (2016). These subdivisions are commonly used by stake-holders to assess impacts of climate variables on economy and society and are the reference adopted by several climate services such as Copernicus for its products (see, e.g. (Brandmueller et al., 2017)). Averaging from the grid-cell size to regional or national scales give us the possibility of both exploring the robustness of our study to coarse-grain and it also allows

to remove part of the variability encountered for precipitation data at grid-level scales caused by model or data issues (Li et al., 2011; Tabari et al., 2016; Herold et al., 2017). In Figures 1-2, NUTS-2 results are represented in panels a,c) and NUTS-0 results in panels b,d). The agreement between the ERA5 and the E-OBSv20.0e data-set largely depends on the regions considered. Overall, the climatologies of snowfall provided by the two datasets have similar ranges although ERA5 tends to overestimate EOBSv20.0e. This confirms that converting precipitation to snowfall using a temperature threshold of $2^\circ$ C is a good option to

retrieve snowfall data from EOBSv20.0e. By analysing the climatology we remark that, at southern latitudes and on the plains, mean and max statistics tend to coincide because the number of snow days per year is limited, i.e. all snowfall is concentrated in one or few events. We can also observe from Figure 2 that coarse graining from NUTS-2 to NUTS-0 level heavily reduces the magnitude of yearly maximum SF.

## 3   Changes in snowfall

We now identify changes in snowfall as differences between average values of both yearly total SF and the maximum yearly SF for two different periods: 1979-1998 and 1999-2018. We subtract the first period from the second, so that positive changes correspond to an increase in snowfall and negative values to a decrease. To check statistical significance of changes, we perform a two sided T-test with confidence level 0.05 (Rushton, 1952). For the yearly total SF of ERA5 (Figure 3a,b) changes are negative for most of Northern, Central and Eastern Europe, whereas near-zero changes are observed in Western Europe. Largest

negative changes are found in correspondence of mountain ranges such as the Alps, the Balkans and Scandinavian Mountains. This decrease in snowfall is significant (green shading) over most of the Northern countries and the British Isles. When coarse grain from NUTS-2 to NUTS-0 level is applied, we can observe that positive changes tend to be averaged out, and significance of negative changes extend to almost all Northern Europe. The picture is similar for the EOBSv20.0e dataset (Figure 3c,d), with mountain regions and Northern countries showing a large decrease of yearly total SF. Positive changes are found in the

Balkans at the NUTS-2 scale, but they are partially averaged out when coarse graining data to the NUTS-0 level (panel d). As for ERA5, negative trends are significant for Iceland, the United Kingdom, Finland, Latvia, Denmark, Italy, France and Turkey.

For the maximum yearly SF, differences observed are generally milder and positive and negative changes are spatially scattered at the NUTS-2 level (Figure 4a,c) for both the datasets. There is however a certain agreement in maximum snowfall

increase over eastern Europe and decrease over western Europe (excluding Spain) among the two data-sets. The NUTS-0 level (Figure 4b,d) provides a more coherent picture with Western Europe characterised by a decrease in maximum snowfall and Eastern Europe where some countries show increasing maximum SF. Significance of changes is low and scattered spatially without a clear geographical coherence. Large differences between the two datasets are found for Switzerland, Greece and Turkey. In ERA5 trends are positive Switzerland and Turkey (negative for Greece) and yield the opposite sign in EOBSv20.0e

dataset. We can justify this difference for Greece and Switzerland using the data at the NUTS-2 level (Figure 4a,c) as they show for regions within those two countries positive and negative differences. At NUTS-0 level (Figure 4b,d) the averaging procedure can therefore provide trends of different signs in EOBSv20.0e and ERA5 datasets, depending on the magnitude of local SF

maxima. The positive trend in ERA5 for Switzerland is in contradiction with several other studies based on snow stations (see e.g. Scherrer et al. (2013); Marty and Blanchet (2012)). For Turkey, the differences between datasets are evident already at the NUTS-2 level and cannot be explained with the spatial averaging. A possible justification comes from the low coherence of the two datasets. This can be checked by computing the symmetric percentage error $\varsigma = (x_{ERA5} - x_{EOBS})/(x_{ERA5} + x_{EOBS})$, where $x$ is the total average snowfall for each region and $\varsigma \in [-1\ 1]$ (Figure 5a,b) and the correlation coefficient $R^2$ between the daily time series of accumulated snowfall of the two datasets (Figure 5c,d). $\varsigma$ shows that ERA5 tends to overestimate snowfall with respect to EOBS over Southern Europe and underestimate over Central Europe and the UK. Entire regions of Turkey, Portugal and Southern Italy show a weak correlation $R^2 \sim 0.3$ between the two datasets, pointing to some problems in the data assimilation possibly due to scarce availability of good quality meteorological data over those regions. Indeed correlation coefficient is larger for Northern European countries that dispose of high spatio-temporal data cover. The error $\varsigma$ reduces and the correlation coefficient $R^2$ increase when considering the NUTS-0 level, since local differences are averaged out.

The ensemble of these analyses suggest that whereas larger confidence can be attributed to a decrease in yearly snowfall over Northern Europe, changes are more uncertain for maximum snowfall. For the maxima, some coherence appears at country scale: negative changes over Western Europe and positive ones in Eastern Europe, specifically for some countries in the Balkans. The difference in the changes for average and maxima suggest a non-trivial relation between the occurrence of extreme snowfalls and global mean warming. In order to explain such changes, we will investigate the role of the atmospheric circulation for the countries in the Baking showing large positive maximum SF changes and with coherent trends between the two data-sets considered.

## 4 Thermodynamic and dynamical analysis for countries with increasing maximum snowfall

The analysis of Figures 2-3 suggests that changes for maximum snowfall are very scattered and even adjacent regions can show changes of different signs. This makes the single region analysis of trends almost meaningless as robust links between SF and large scale fields are likely to be very weak. We therefore focus on the NUTS-0 positive trends of ERA5. We decide to use ERA5 because snowfalls are produced by the model underlying the reanalysis and naturally associated with coherent circulation patterns. We discard the E-OBSv20.0e data-set as it does not contain other atmospheric variables that could help in tracking the atmospheric thermodynamics or the large-scale atmospheric circulation. We identify the 3 countries showing coherent positive changes for both datasets, namely Albania (AL), Montenegro (ME), Bosnia (BA). The countries selected in the Balkans are interesting because they also show positive or zero trends for total yearly SF in the ERA5 data-set. In this section we focus on the intensity of positive changes regardless of their significance. As pointed out by Altman and Krzywinski (2017), statistical testing based on $p$values presents several limitations, and can produce misleading results even in designed experiments. Here, we privilege the physical complexity of the phenomenon, as information about pure statistical significance has already been discussed in the previous section. In Figure 6a) we show the box-plots of the yearly snowfall maxima organized in the two different periods (1979-1998 and 1999-2018) for the 3 regions identified. Boxplots provide more detailed information on the

nature of changes: whereas for Bosnia and Montenegro the bulk of the distribution shift towards larger values in the second period, for Albania increase of maximum snowfall is mostly due to two outliers, which occur at the end of the second period. For Albania and Montenegro the variability also increased in 1999-2018, while it is stationary for Bosnia.

The analysis presented in Figure 6b) aims at identifying possible seasonal variations of extreme snowfalls. In the polar plot, the radius corresponds to the average magnitude and the angle to the date of the year of SF maxima. For the countries considered, there is a tendency to observe heavy snowfall later in the winter season, although the shift is rather modest.

To understand the nature of the changes, we will analyse the synoptic environment associated with snowfall events from both a thermodynamic and dynamical point of view, analysing indicators of stability of the atmosphere as well as circulation patterns (weather regimes) during the events.

## 4.1 Thermodynamic changes

The first hypothesis to explain the occurrence of increasing heavy snowfalls despite the current global warming trends is that starker sea surface - troposphere temperature contrasts might enhance moisture uptake in combination with reduced stability, that trigger ascending motions and local convergence during snowfall events. This possibility has been explored in other studies by looking at atmospheric stability and air-sea interaction during cold-air outbreaks, albeit for different regions (Papritz and Spengler, 2015, 2017; Czaja et al., 2019). Furthermore, in an event-based study of cold and snowy spells over Italy, D'Errico et al. (2019) link the recent enhancement in snowfalls on the Adriatic regions to the increase of convective precipitations from the Mediterranean sea, which is warming faster than the oceans at same latitudes because of its closed geometry (Gualdi et al., 2013). To explore this possibility, we look at changes in the stability of the flow during the maximum snowfalls using the 2-meters temperatures (t2m, boxplot in Figure 6c) and the Convective Available Potential Energy (CAPE, boxplot in Figure 6d). The choice of CAPE as indicator of stability during snowfall is motivated by previous works (e.g. Schultz (1999); Olsson et al. (2017)) where snowfall extremes were co-associated with the occurrence of high CAPE values. The boxplots in Figure 6c-d) show the spatial regional average of t2m and CAPE during the days of the maximum snowfalls. The analysis for temperature (Figure 6c) suggests that maximum snowfalls tend to be associated with temperatures above the freezing point in the recent period and a reduced variability with respect to the first period. It is not unusual to observe snowfall with ambient temperatures up to $6°$ C (see, e.g. Steinacker (1983)) and more recent studies show that snow and rain can coexist even up to $13.3°$ C (Wen et al., 2013). The analysis for the local CAPE (Figure 6d) is not very informative for two reasons: i) the distribution is highly non-Gaussian, it includes zeros and presents several outliers, ii) convective precipitations can originate in nearby regions and being transported.

In order to explore this possibility, from bloxpots in Figure 6 we move to the fields analysis in Figure 7 for CAPE and Figure 8 for t2m. From Figure 7 we remark that heavy snowfalls for the 3 countries under examination are generally associated with large values of CAPE on the Adriatic sea. For both periods, the absolute values of CAPE reached during these events (Figure7a,b,d,e,g,h) are consistent with the range found by Olsson et al. (2017) for the enhancement of snowfall by sea-air interactions. When looking at differences ($\Delta$CAPE) between the warmer (1999-2018) and colder (1979-1998) period

(Figure 7c,f,i), we remark that there is an increase of CAPE (shaded regions indicate that at least 2/3 of the anomalies yield the same sign) suggesting that convective instability can be an important factor in Adriatic regions to trigger heavy precipitations on the Balkans. Furthermore, snowfall extremes tend to occur at or near the freezing point in both colder and warmer climates (O'Gorman, 2014). Figure 8 indicates that this is the case for both the periods considered and that the local temperature difference ($\Delta$t2m) between the 1979-1998 and 1999-2018 periods is small (see also boxplots in Figure 6c). The local temper-

ature is important as it determines the maximum atmospheric moisture content and thus the thermodynamic component of the snowfall amount. While the warming of the Mediterranean Sea during these events (Figure 8c,f,i) favors evaporation, part of the excess moisture to precipitate out during the transport to this location, possibly reducing the thermodynamic enhancement of the snowfall.

## 4.2 Dynamical Analysis

Since thermodynamic effects alone cannot explain the positive changes in maximum snowfall, we also investigate the role of the atmospheric circulation as a driver of those changes. For other regions of the world, this kind of analysis has provided evidences of a prominent role of atmospheric circulation on the variability of extreme snowfall events (see e.g. Kawase et al. (2016) for Japan, Lute et al. (2015) for Andorra or Guan et al. (2010) for USA). For the countries examined in the present study, the motivation for such analysis comes from the evidence that recent decades show a reduction in the abundance of zonal

patterns (Guan et al., 2010).

We first present the sea-level pressure (msl) fields averaged during maxima of snowfall for both the periods in Figure9a,b,d,e,g,h) and their differences $\Delta$msl in Figure9c,f,i). For the three countries considered, cyclonic patterns over the Thyrrenian and the Adriatic sea can be identified for both the periods considered. Following pressure isobars, the flow is advected from sea-to land. In 1999-2018, cyclonic conditions further strengthened in the Balkans with negative $\Delta$msl anomalies over Eastern Eu-

245 rope, suggesting that, in the recent period, moisture advection from the Mediterranean sea to the Balkans has favored snow accumulations on the countries examined. These cyclonic patterns can originate from two different conditions: local cycloge-nesis on the Adriatic sea driven by cold air intrusions from Siberia (Bisci et al., 2012) or cyclogenesis on the Thyrrenian sea (Genoa Lows) driven by polar air flowing through the Rhone Valley (Spreitzhofer, 2000). The $\Delta$msl analysis (Figure 9c,f,i) shows the reinforcement of a Eastern Mediterranean cyclonic pattern in the second period. The isobars point to southerly

winds which favor uptake moisture from the Mediterranean Sea. Thus, the increase of CAPE over the Adriatic shown in Fig-ure 7c,f,i), together with the reinforcement of the pressure lows over Italy, could determine the increase in extreme snowfalls in these countries.

Following the approaches of Vautard (1990) and Michelangeli et al. (1995), we now analyse the shifts in daily weather regimes associated with extreme snowfalls. Weather regime search is performed by using the dynamical systems indicators

introduced in Faranda et al. (2017) and using the sea-level pressure fields for the same domain specified in that study, namely latitudes 22.5N-70N, and longitudes 80W-50E. The technique presented in Faranda et al. (2017) allows to determine five pos-sible regimes: North Atlantic Oscillation positive (NAO+) and negative (NAO-) phases, Blocking (BLO), Atlantic Ridge (AR) and non-attributable patterns (N/A). These patterns have been previously identified in many studies over this domain (see e.g.

Vautard (1990)). The patterns obtained for ERA5 do not differ significantly from those shown in Figure 2 in Faranda et al. (2017). Results are shown in Figure 10 for the countries examined and show a prevalence of BLO and AR patterns during extreme snowfall events. These patterns (see e.g. D'Errico et al. (2019) and references therein) favor meridional movements of air masses and therefore the intrusion of polar air to Mediterranean latitudes. It is remarkable that, for all the three countries considered, the second period is characterised by an increase of Atlantic Ridge and Blocking patterns. These pattern consists of high pressure over Western or Northern Europe, favoring dry conditions over Western Mediterranean areas, and low pressure over Eastern Europe, triggering cyclogenesis on the Mediterranean and favoring the intrusion of cold air from Siberia (Raymond et al., 2018).

The previous analysis shows that the weather regimes shift is an important factor determining changes in extreme snowfall. However, the statistics presented in Figure 10 is limited by data availability. We therefore extend this analysis by performing an analogs search for the 5% closest sea-level pressure fields (according to the Euclidean distance) to those presented in Figure9a,b,d,e,g,h,j,k) (Yiou et al., 2013). Note that the results do not depend on the threshold used for the selection of analogs in the range 0.25% to 5%. For each of those fields, the analogs search is performed in all the dataset (1979-2018). We then plot in Figure 11 the number of analogs per year. A linear fit is applied to data. Besides Bosnia (Figure 11c), for which the increasing trend in the number of Analogs is significant (5% level), for the other countries considered, trends are not significant. Furthermore no clear differences appear when searching analogs for 1979-1998 or 1999-2018 fields associated with extreme snowfalls. This means that the changes in circulation patterns associated with extremes are specific to those events, and seem not follow some general trends of the atmospheric circulation, thus suggesting a competition between thermodynamic and dynamical factors in their occurrence. The rationale for explaining the changes is the following: AR and BLO patterns occur with the same frequency in winter but they are associated with deeper cyclogenesis in the Adriatic or Thyrrenian sea in the recent period. These cyclones find warmer surfaces and large availability of humidity and CAPE, thus producing large snowfall amounts, enhanced by Stau effect on the Balkans topography. This mechanism is similar to the one described for Japan in Kawase et al. (2016) and for Italy in D'Errico et al. (2019).

## 5   Conclusions

We have analysed recent changes in yearly total and maximum snowfall from ERA5 reanalysis and the E-OBSv20.0e data-sets. We have identified a robust signal in the general decrease in the yearly total snowfall, in particular for Northern and Western Europe. For snowfall maxima, changes are more contrasted: negative changes persist over Western Europe, but in the proximity of the Mediterranean Sea we have identified a certain number of countries showing positive changes. We have focused our efforts in understanding the positive trends for maximum snowfalls in some countries the Balkans which showed consistent positive trends for both the datasets considered. The thermodynamic analysis of atmospheric stability and 2-meters temperatures suggest that during recent heavy snowfall events the instability increases and convection is favored, an effect that could be linked to climate change (Ye et al., 1998). This can however be contrasted by the fact that excess moisture could precipitate out during the transport to the snowfall location due to temperatures close to freezing points. The thermodynamic analysis has been

completed by an analysis of the atmospheric circulation patterns associated with extreme snowfall over these countries. Results show an enhancement of local cyclonic patterns and a tendency to observe Atlantic Ridge patterns associated with extreme snowfalls in recent times. Even though this could suggest a relation between our finding and the arctic amplification caused by climate change (Vavrus et al., 2017), we stress that the length of the data-sets used is too short to attribute these changes to climate change and that they could be produced by the inter-decadal variability of the atmospheric circulation. Furthermore, the analogs analysis carried out in Section 4 did not show any particular trends in analogs for all the countries considered but Bosnia. Recent studies on whether these patterns are due to low-frequency variability of the Atlantic circulation or to climate change are debated (see, e.g., the discussion in Screen (2017)).

To summarize our findings, there is an interplay of circulation and thermodynamic factors to explain the observed trends in maximum snowfalls on the Balkans: the analysis of CAPE shows that large values of this quantity are associated with heavy snowfalls in the selected countries. CAPE values of $70\ \mathrm{JKg^{-1}}$ are enough to trigger convection during winter time and enhance snowfall precipitations (Olsson et al., 2017). Furthermore, for all countries analysed, the isobars associated with the cyclonic conditions embedded in Atlantic ridge patterns indicate winds blowing from sea to land, thus favoring the advection of moisture and the formation of convective precipitation. In addition, the three countries analysed are characterised by mountain ranges that, in presence of sea-to-land flow associated with extratropical cyclones, favors the Stau effect on precipitations (Bica et al., 2007). Both thermodynamics and dynamics effect seem therefore to contribute to observed trends, although it is difficult to understand which factor prevails. Only the thermodynamic components of increasing instability can be linked to climate change (Ye et al., 1998). Although winter total precipitations in future climate scenarios is expected to increase over Europe (Santos et al., 2016), global and regional warming is projected to reduce average and extreme snowfall precipitations at least in Central and Western Europe (de Vries et al., 2014). In the same study, de Vries et al. (2014) find that positive trends in snowfalls could still be observed for high mountain areas (Alps and Scandinavia) in warmer climates. This seems coherent with the results found for Japan by Kawase et al. (2016) and in the present study for the Balkans.

This study comes with some caveats. First of all, the changes (especially those on the maxima) depend on the dataset chosen. Here we have focused on consistent trends between EOBSv20.0e and ERA5 and then used ERA5 for the analyses because of the consistent representation of snowfalls with the atmospheric circulation. The lack of longer and highly resolved data-sets for snowfall is a strong limitation and it adds up to the intrinsic difficulty of simulating snowfalls due to their highly non-linear behavior and the fact they involve phase transitions. In addition, we have not considered the effects on the trends of lower frequency variability mechanisms. There are sub-seasonal to seasonal conditions that can trigger snowy waves over Europe by modifying winter atmospheric circulation patterns: the role of stratospheric warming, the magnitude of snow cover on Siberia and in the Arctic region could be taken into account in future research on this topic, e.g. by following the approaches of Handorf et al. (2015, 2017) and Mori et al. (2019). At smaller scales, where convection is important, further studies could be based on searching the origin, transport pathways, and thermodynamic evolution of air masses involved in heavy snowfall

episodes, via novel methodologies based on tracking trajectories of air masses as those introduced in Papritz and Spengler (2017), and by using convection permitting models to study sea-air-snow interactions (Bartolini, 2019).

## 6 Acknowledgments

The author wishes to thank F Pons, S Fromang, P Yiou, M Vrac for the discussions and S Thao for the help with the ERA5 data-set. The author acknowledge the support of the INSU-CNRS-LEFE-MANU grant (project DINCLIC), as well as EUPHEME and DAMA. This work is supported by the ANR-TERC grant BOREAS The author acknowledges two anonymous reviewers as well as the editorial board of WCD for useful comments on the manuscript.

*Competing interests.* The author declares no competing interest.

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

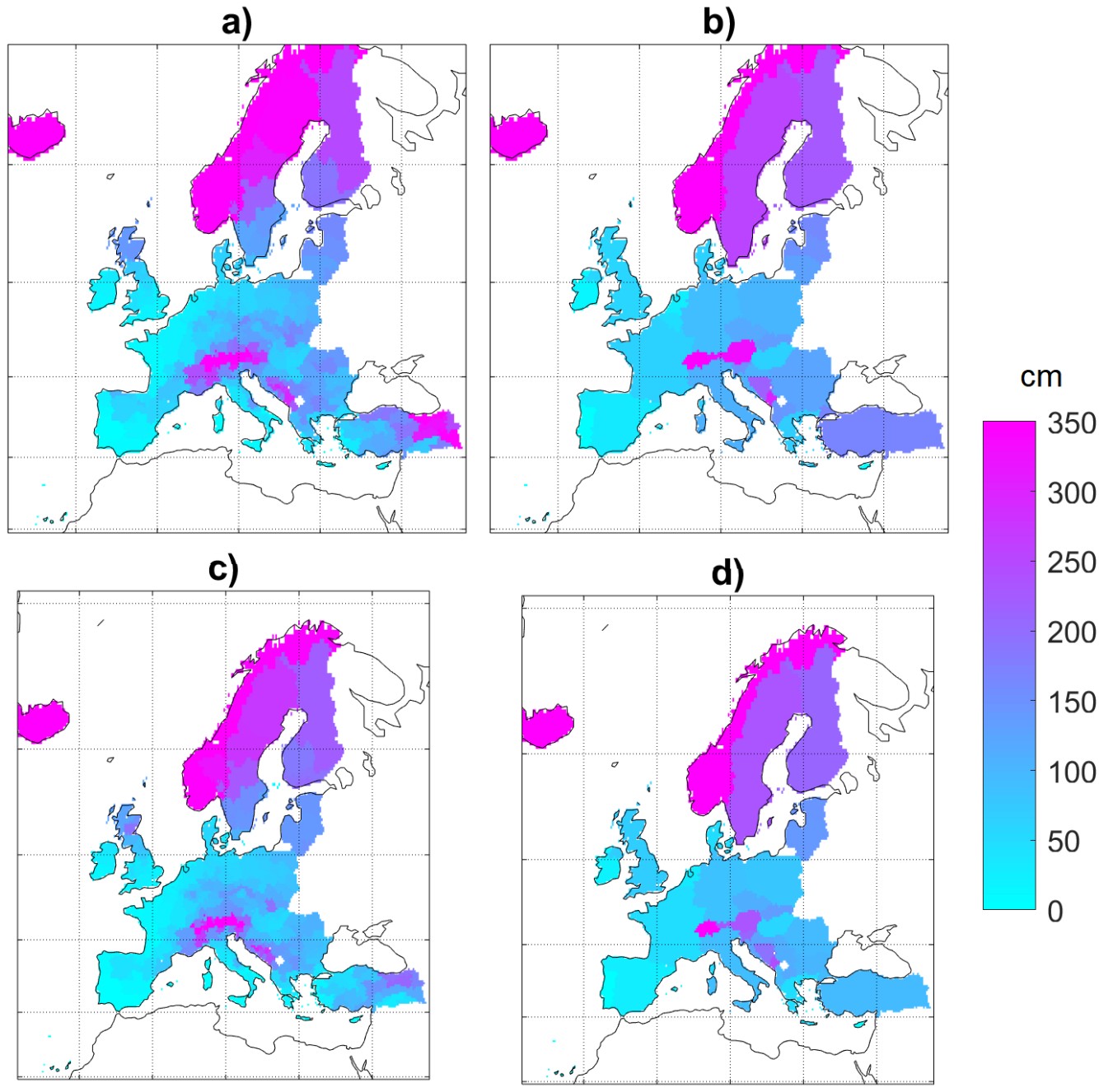

**Figure 1.** Yearly total snowfall SF (average 1979-2018) for the ERA5 (a,b) and the E-OBSv20.0e (c,d) data-sets. a,c) NUTS-0 level, b,d) NUTS-2 level.

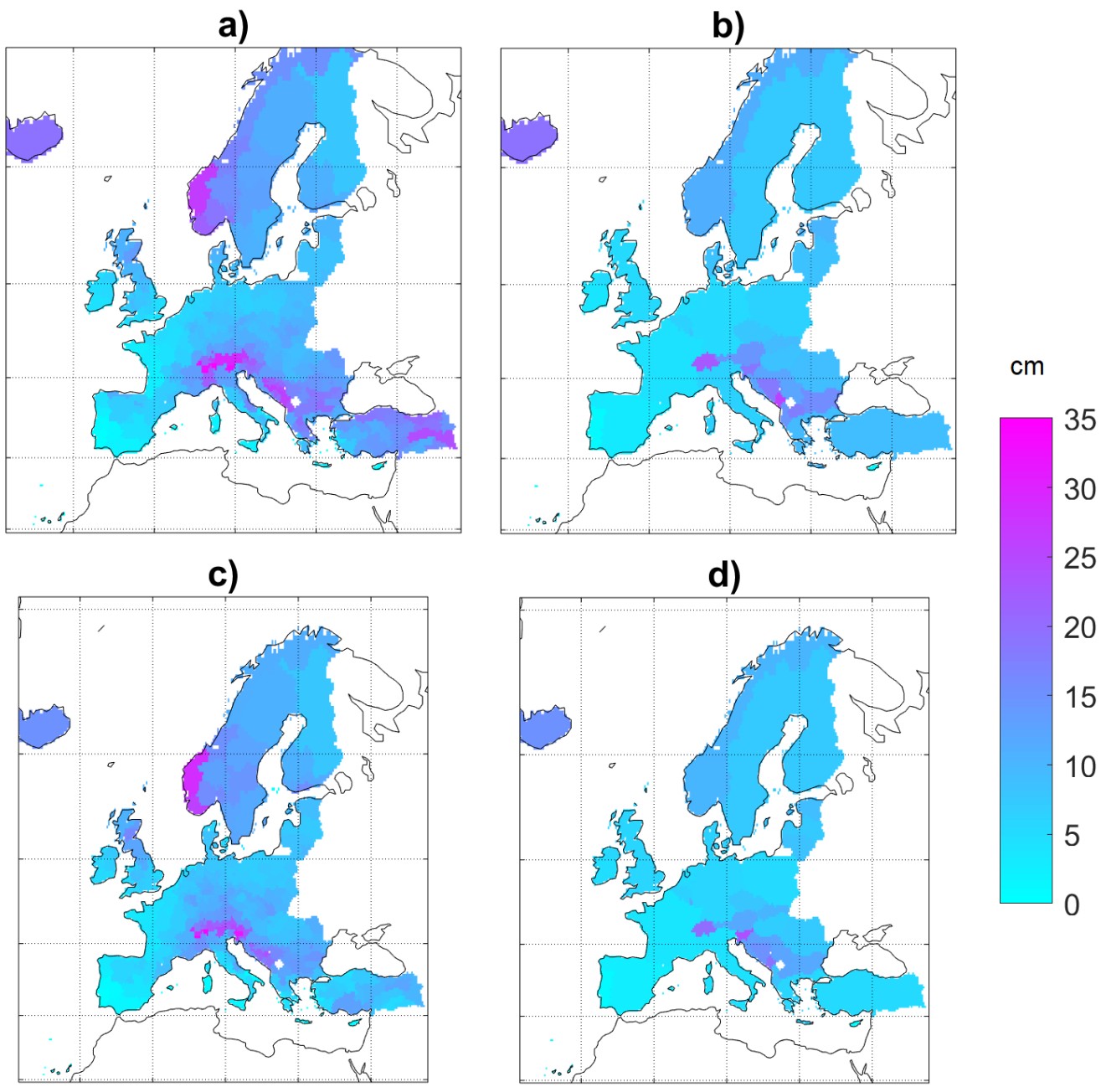

**Figure 2.** Maximum yearly snowfall SF (average 1979-2018) for the ERA5 (a,b) and the E-OBSv20.0e (c,d) data-sets. a,c) NUTS-0 level, b,d) NUTS-2 level.

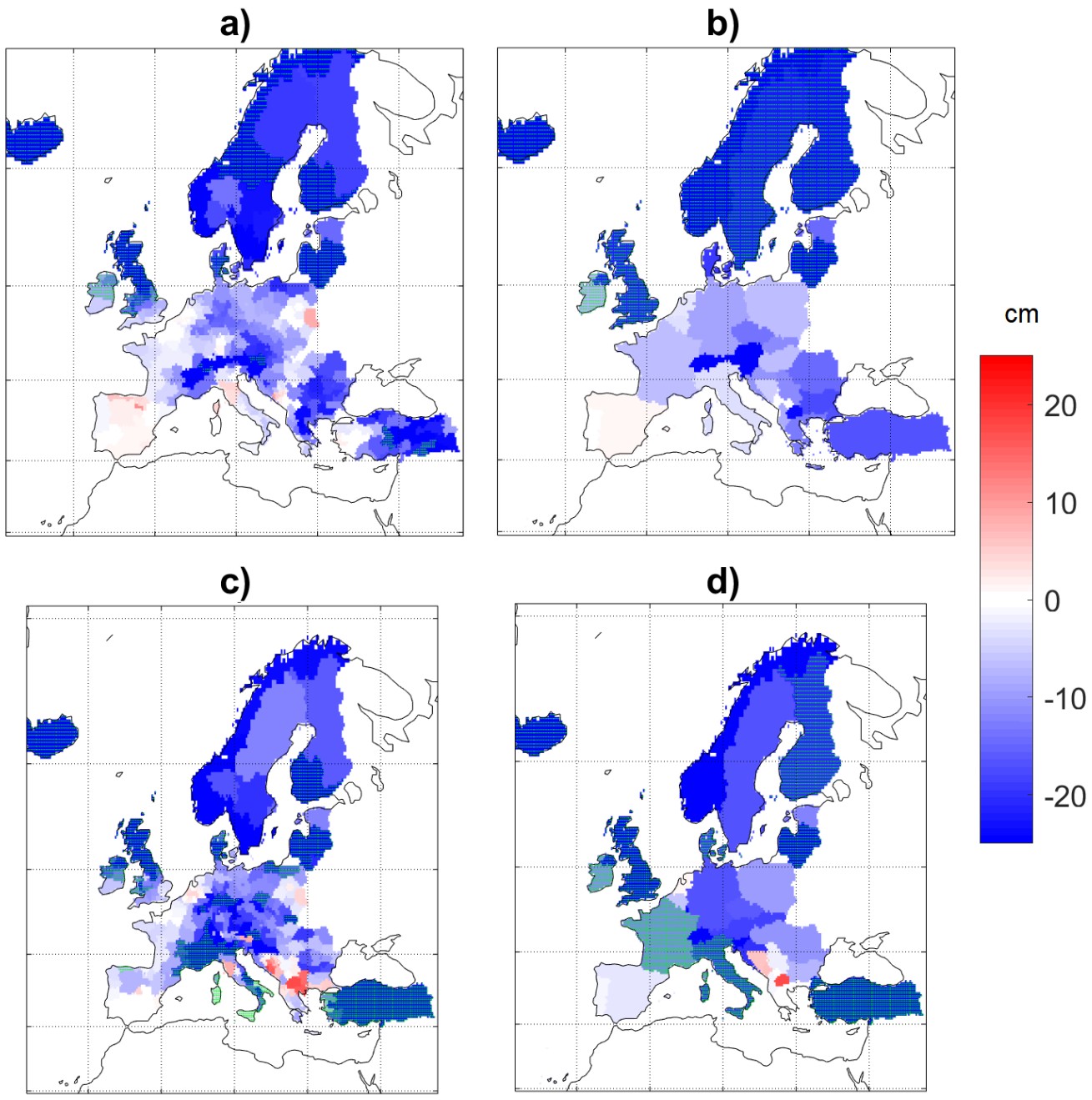

**Figure 3.** Differences in yearly total snowfall (SF) for two periods (average 1979-1998 subtracted from average 1999-2018) for the ERA5 (a,b), E-OBSv20.0e (c,d) data-sets. a,c) NUTS-0 level, b,d) NUTS-2 level. Significant differences are shown in shaded green (two-sided T-test, 5% confidence level) .

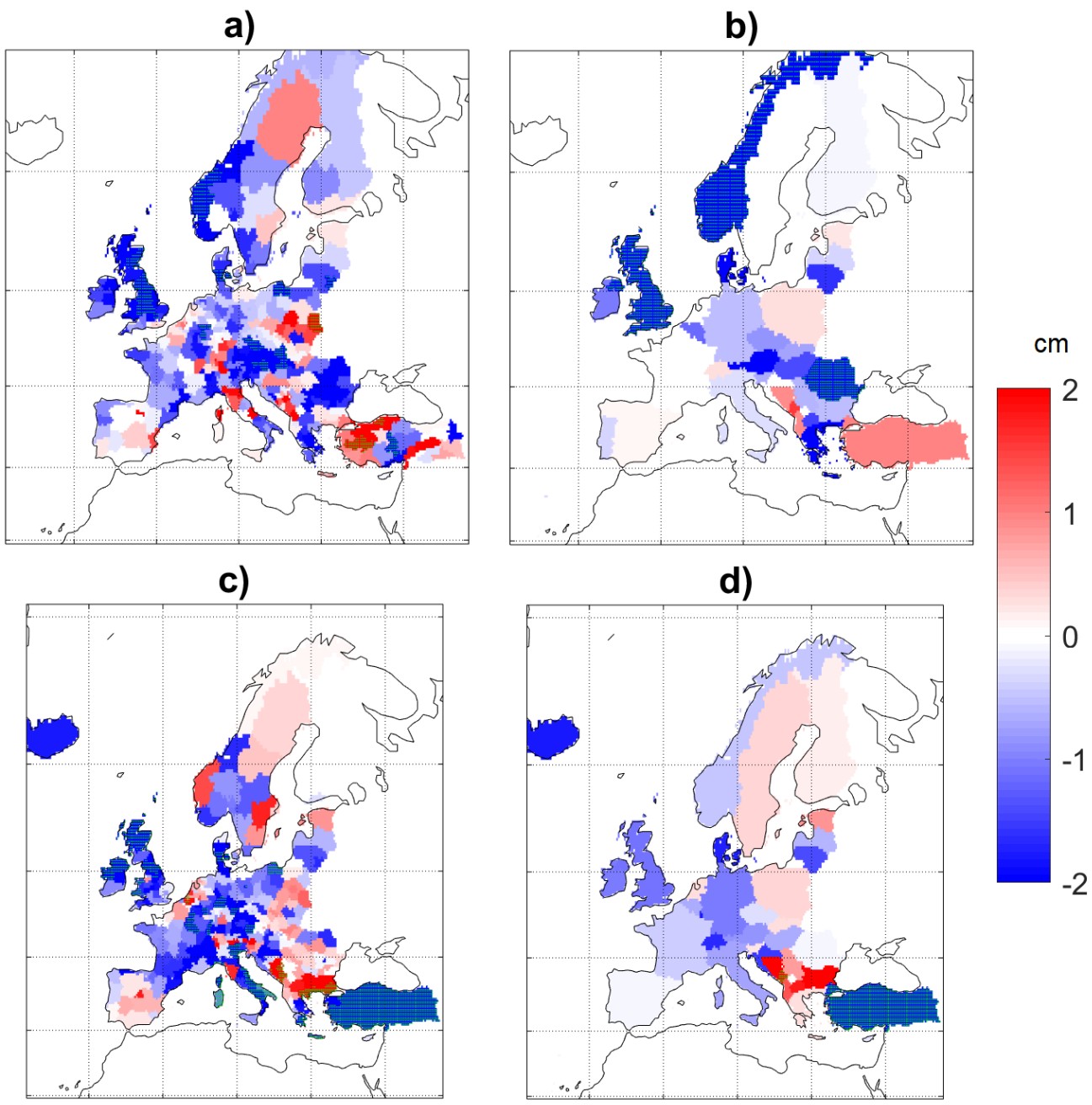

**Figure 4.** Differences in maximum yearly snowfall (SF) for two periods (average 1979-1998 subtracted from average 1999-2018) for the ERA5 (a,b), E-OBSv20.0e (c,d) data-sets. a,c) NUTS-0 level, b,d) NUTS-2 level. Significant differences are shown in shaded green (two-sided T-test, 5% confidence level) .

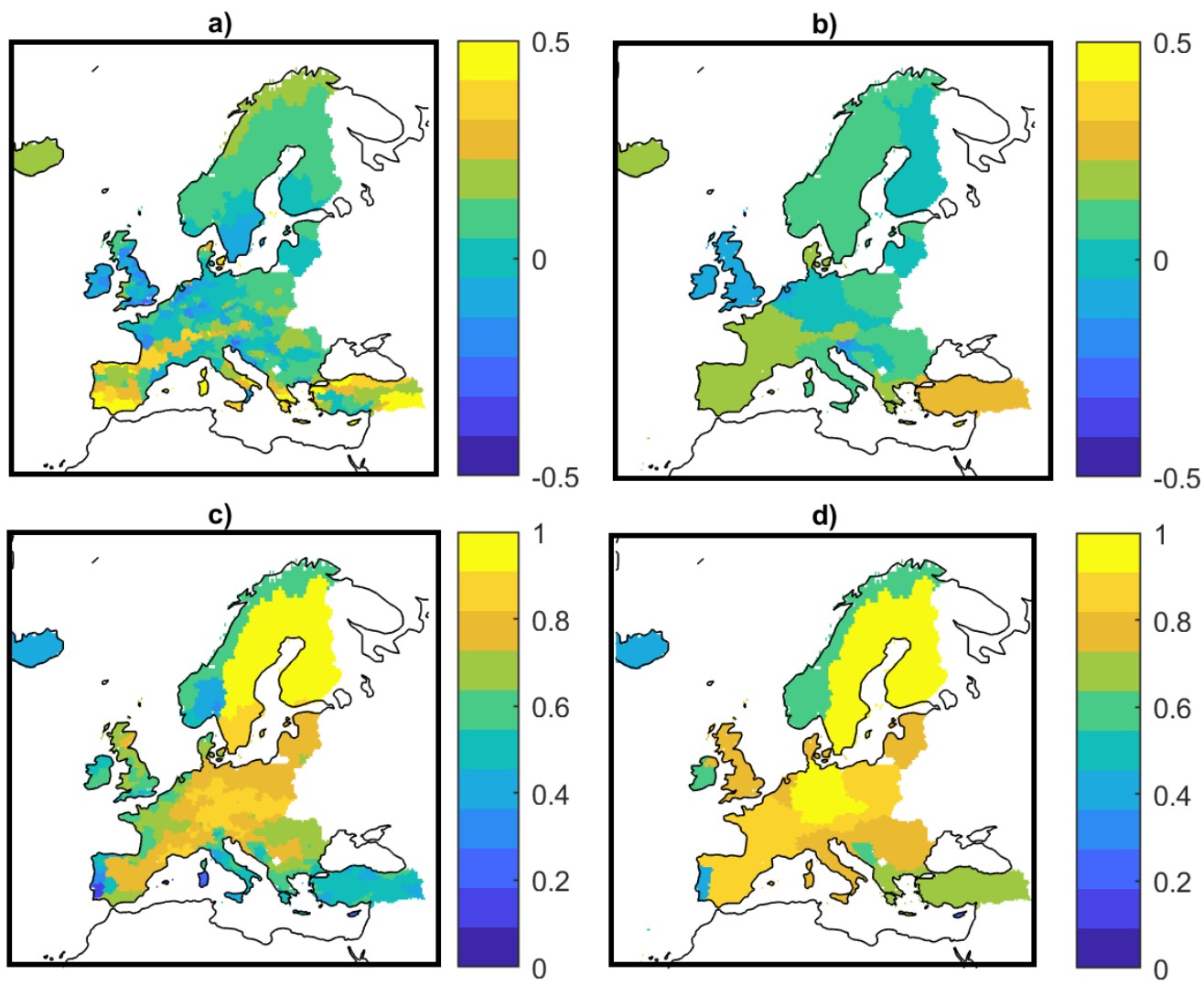

**Figure 5.** a,b) Symmetric percentage error ς on average snowfall c,d) Correlation coefficient $R^2$ for the SF daily snowfall time-series for ERA5 and E-OBSv20.0e. a,c) NUTS-2 level, b,d) NUTS-0 level

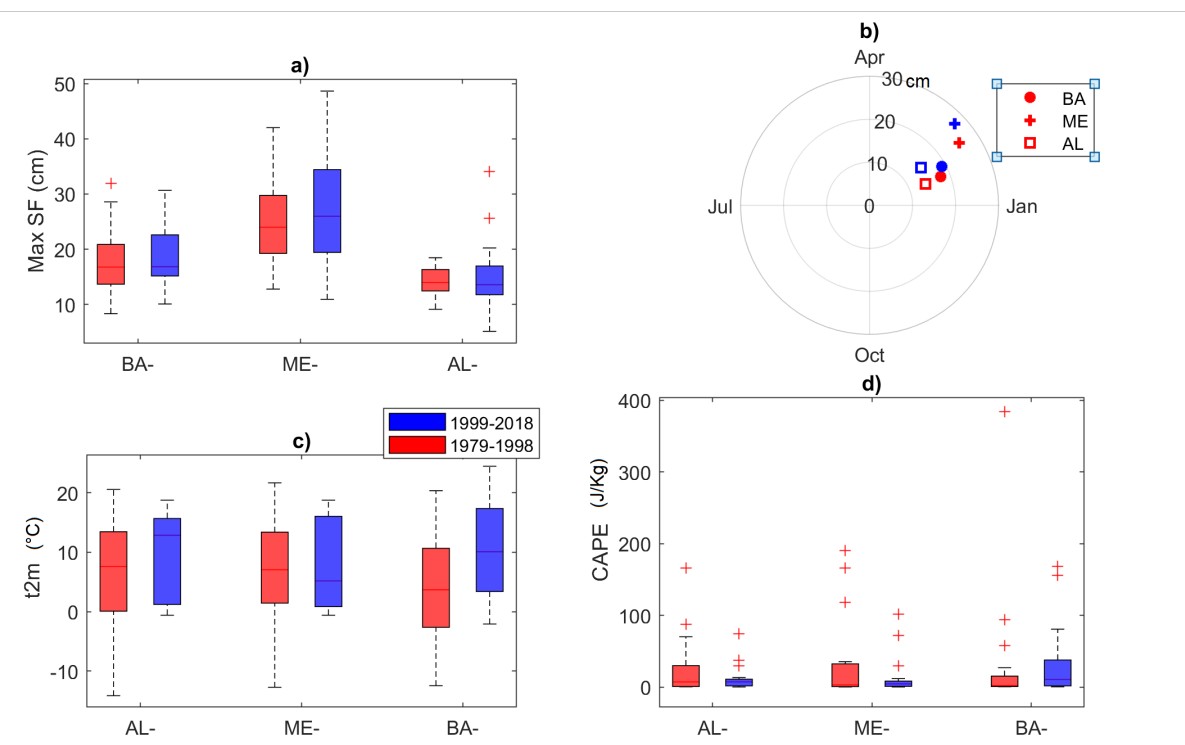

**Figure 6.** a) Boxplots of the maxima yearly snowfall (SF) for two periods 1979-1998 (red) and 1999-2018 (blue). b) Seasonal analysis for maximum yearly snowfall SF. The polar plots show average maxima yearly snowfall for the two periods (different colors). Each symbol corresponds to a different country. The angle corresponds to a date of the year in counterclockwise sense. Albania (AL); Montenegro (ME), Bosnia (BA). Boxplot of 2-meters temperatures t2m (c) and CAPE (d) observed during extreme snowfall events. For boxplots, on each box, the central mark indicates the median, and the bottom and top edges of the box indicate the 25th and 75th percentiles, respectively. The whiskers extend to the most extreme data points and the outliers are plotted individually using the '+' symbol.

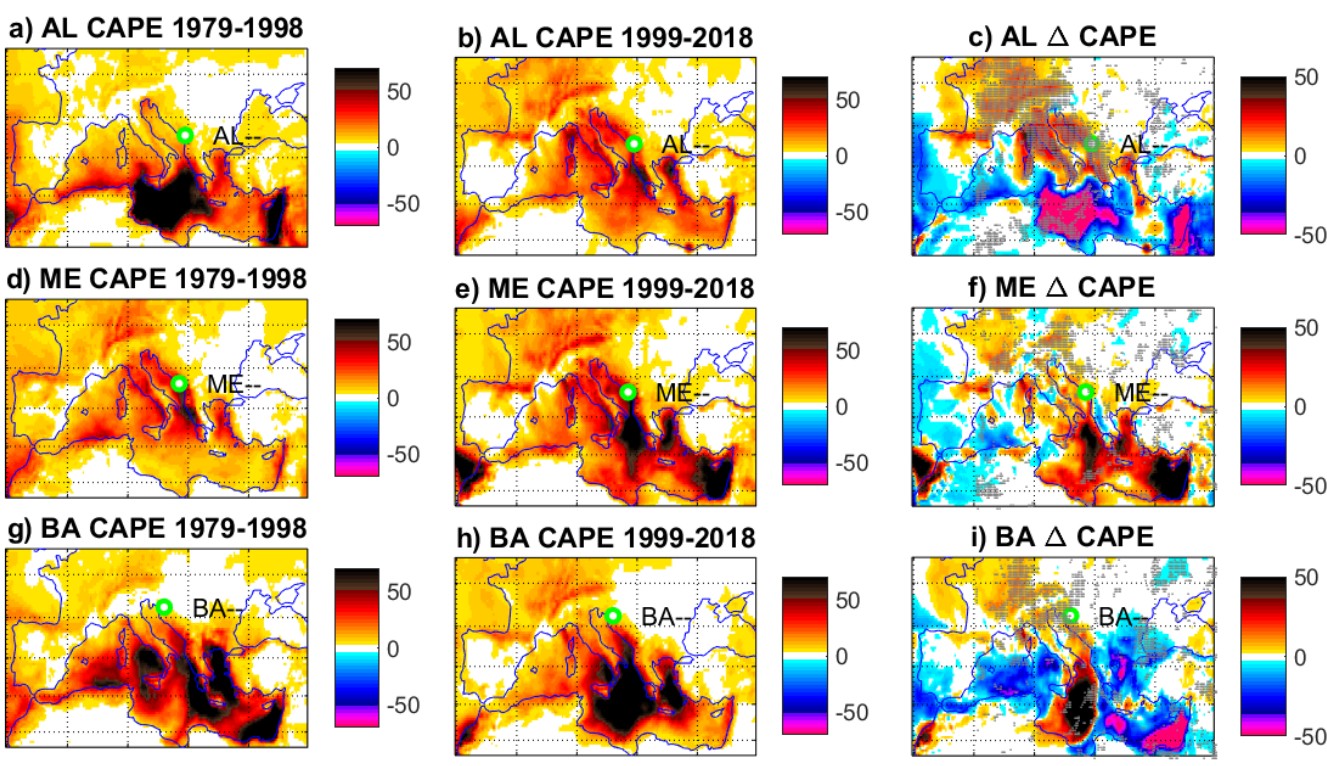

**Figure 7.** Average of convective available potential energy (CAPE) fields ( J Kg$^{-1}$) during days of yearly maximum snowfall for the periods 1979-1998 (a,d,g) and 1999-2018 (b,e,h). Panels c,f,i show the differences between the second and the first periods $\Delta$CAPE. a,b,c) Albania (AL); d,e,f) Montenegro (ME), g,h,i) Bosnia (BA). Green circles show the location of the most northwestern point of each country. Shaded areas represent grid points where at least 2/3 of the anomalies have the same sign.

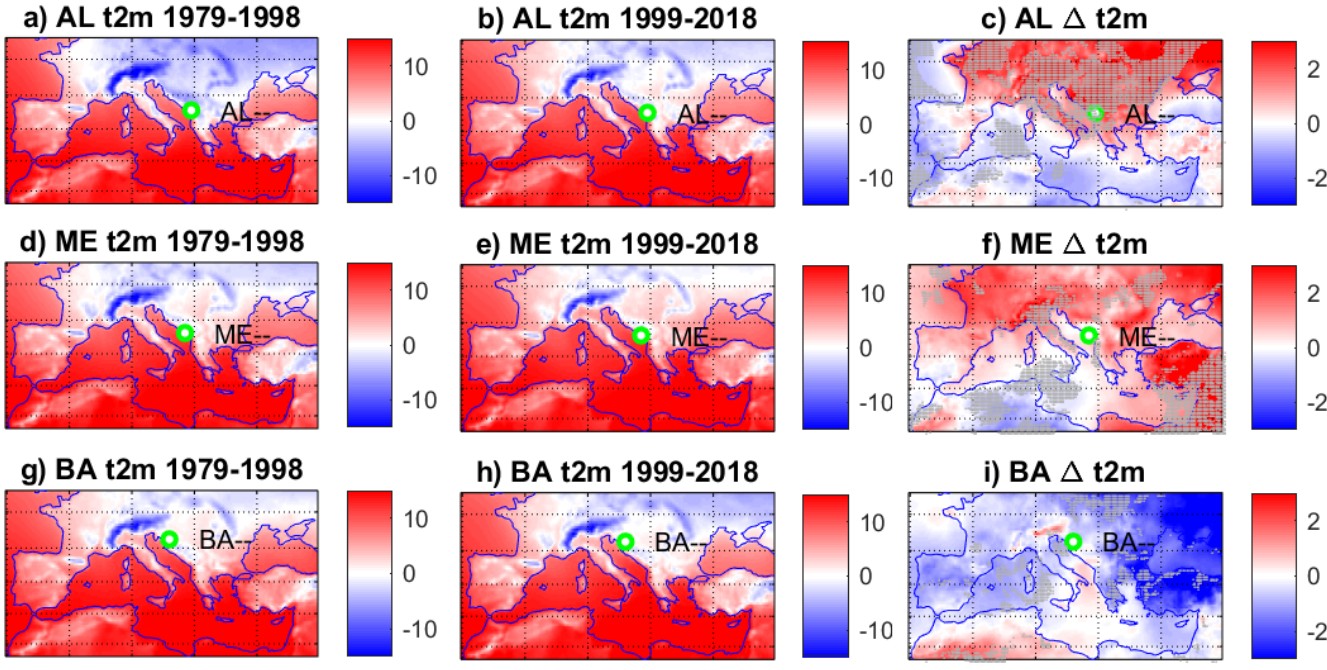

**Figure 8.** Average of 2-meters temperature fields (°C) during days of yearly maximum snowfall for the periods 1979-1998 (a,d,g) and 1999-2018 (b,e,h). Panels c,f,i show the differences between the second and the first periods Δt2m. a,b,c) Albania (AL); d,e,f) Montenegro (ME), g,h,i) Bosnia (BA). Green circles show the location of the most northwestern point of each country. Shaded areas represent grid points where at least 2/3 of the anomalies have the same sign.

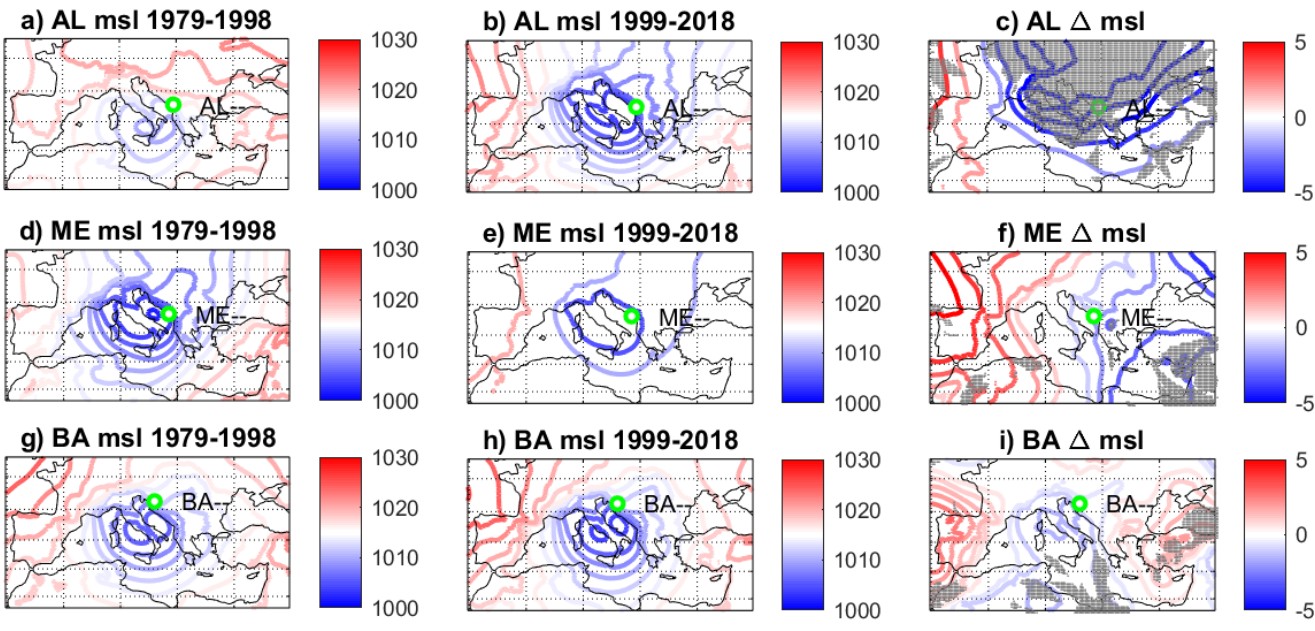

**Figure 9.** Average of sea-level pressure (msl) fields (hPa) during days of yearly maximum snowfall for the periods 1979-1998 (a,d,g) and 1999-2018 (b,e,h). Panels c,f,i show the differences between the second and the first periods Δmsl. a,b,c) Albania (AL); d,e,f) Montenegro (ME), g,h,i) Bosnia (BA). Green circles show the location of the most northwestern point of each country. Shaded areas represent grid points where at least 2/3 of the anomalies have the same sign.

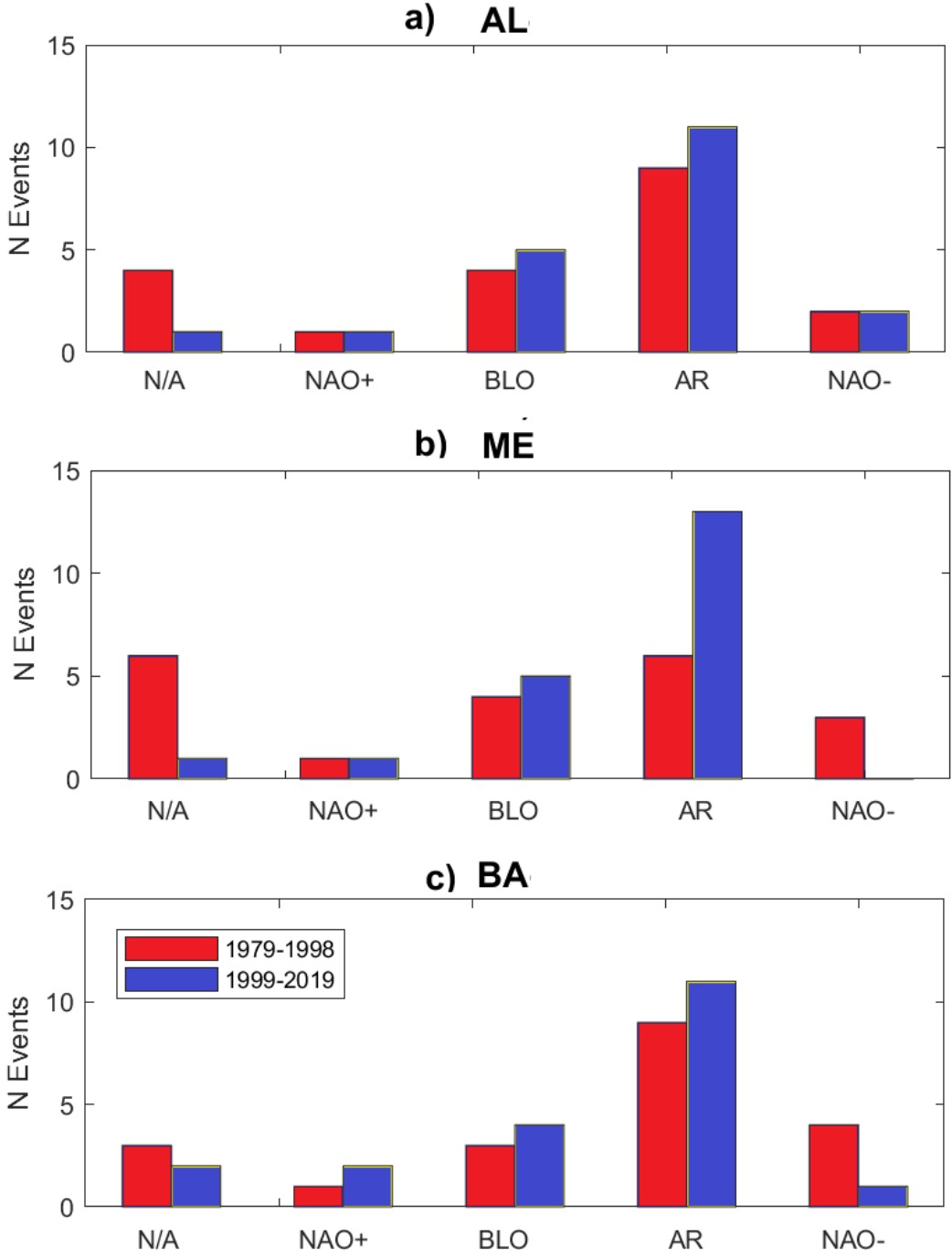

**Figure 10.** Histograms of weather regimes during the days of maximum snowfalls. N/A: non attributable, NAO+: North Atlantic Oscillation positive phase, BLO: blocking, AR: Atlantic Ridge, NAO-: North Atlantic Oscillation negative phase, a) Albania (AL); b) Montenegro (ME), c) Bosnia (BA). Red bars correspond to 1979-1998 and blue ones to 1999-2018.

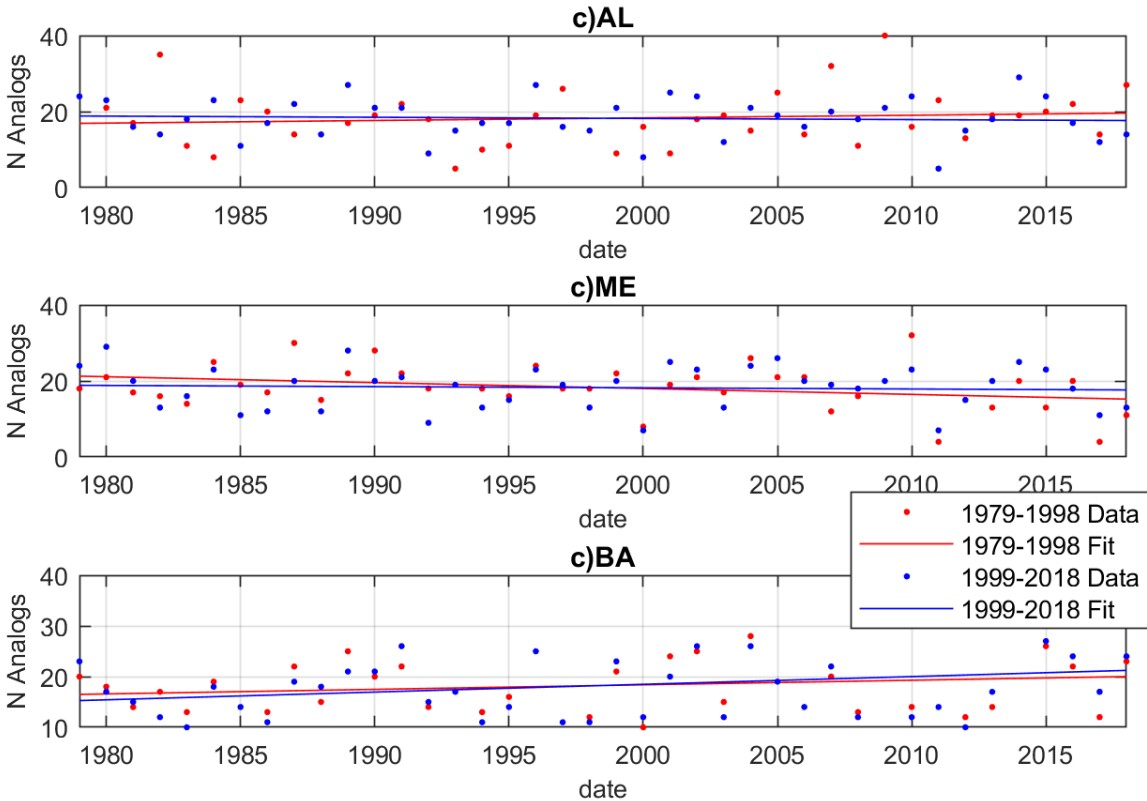

**Figure 11.** Number of analogs per year of the average sea-level pressure fields during days of yearly maximum snowfall for the periods 1979-1998 and 1999-2018. a) Albania (AL); b) Montenegro (ME), c) Bosnia (BA). Red corresponds to 1979-1998 and blue to 1999-2018.