# Peer review of "An attempt to explain recent changes in European snowfall extremes"

_Weather and Climate Dynamics, 2019_

## Referee Comment (RC1) · Anonymous Referee #1 · 16 Jan 2020

Comments on " An attempt to explain recent trends in European snowfall extremes" by Dr. Davide Faranda submitted to WCD.

General comments: In this study, the author investigates recent trends in yearly total snow depth and maximum snow depth in the European region, and for the latter discusses the relationship between the trend and atmospheric circulation and global warming. The reviewer agrees that the manuscript contains a lot of scientific interests to be published since the author focuses on a counterintuitive result the increasing maximum snow-depth trend under global warming. The author tried to understand the relationship between the result and change of atmospheric circulation. However, the relationship or causality would be not fully discussed to be published in this manuscript, and in the current status it seems not suitable to the scope of the WCD journal, be-

[Printer-friendly version](}

[Discussion paper](}

cause the current manuscript contains less investigation on the atmospheric dynamics that causes trends of yearly total and maximum snow-depth. Therefore, I would recommend to resubmit this paper after substantial revision for discussion on the atmospheric dynamics, or the author may address to more elaborate on an observational study such as the comparison with in-situ observations and the ERA5 reanalysis datasets.

Specific comments: The conclusions described in Abstract and Conclusions seem not be supported by the results in Sections 2-4. It looks to me only the result that support the conclusion is "This suggests a non-trivial relation between the occurrence of extreme snowfalls, global mean warming and the internal, long-term variability of the atmospheric circulation" (L136-137). Discussions about atmospheric circulations are too few and thus it is difficult to conclude that "the subtle effects of atmospheric circulation in driving extreme events and the non-trivial relation with global warming: a warmer Mediterranean Sea may enhance convective precipitation in winter-time and trigger heavy snowfalls" (L7-9). At least, there is no figures and discussions on specific humidity, climatological temperature that can determine whether snowfall or rainfall, and sea surface temperature and its related surface fluxes (latent and sensible) on Mediterranean Sea. Also, the reviewer is dissatisfied that the ambiguity of which scale of the atmospheric circulations the author focused on: the synoptic scale or the low-frequency variability? This point was difficult to be understood in Introduction and Section 4.

Another concern is that the author compared the daily composite fields of the period 1979-1998 with those of 1999-2018 (Figs. 6-9). If my understanding is correct, this comparison is the average of 20 daily fields versus that of 20 fields. It seems to me that the number of composite fields is not enough to discuss the daily atmospheric fields, since the daily fields can emphasize synoptic disturbances such as locations of extratropical cyclones. Thus we may need more larger number of daily fields to be composited, or focus on longer timescale fields for low-frequency variability (e.g., Nakamura et al. 1997). (Nakamura et al. 1997: "The Role of High- and Low-Frequency

Dynamics in Blocking Formation ", Monthly Weather Review)

In addition, there would be less discussion on the relationship between atmospheric circulation and global warming. For example, could you compare the increasing/decreasing snow-depth trends with estimation of the Clausius-Clapeyron relationship?

Instead, the author could focus on the observational part. I am not familiar with observation research, yet it would be valuable and novel to compare the ERA5 snow estimations with observations. It will provide useful information for reanalyses that are crucially important for weather and climate researches.

It would be helpful to refer to Kawase et al. (2016) who investigated future changes of averaged (yearly total) and extreme (maximum) snowfall events over Japan (East Asian regions), and their results seem partly consistent with your results here. Also you can find Steenburgh and Nakai (2020) for some reviews of snowfall over Japan. (Kawase et al. 2016: Enhancement of heavy daily snowfall in central Japan due to global warming as projected by large ensemble of regional climate simulations, Climatic Change. Steenburgh and Nakai 2020: Perspectives on sea- and lake-effect precipitation from Japan's "Gosetsu Chitai", Bulletin of the American Meteorological Society)

Technical corrections: -L143: What is the "ERA5 data per NUTS2"? Please describe. -L153: "hep" => "help". - What is "the block-maxima procedure"? Please explain.

---

## Referee Comment (RC2) · Anonymous Referee #2 · 6 Feb 2020

**General comments:**

The article investigates extreme snow depth trends in Europe in the last 40 years and attempts to explain these trends in light of global warming and changes in atmospheric circulation. I find the topic interesting and definitively of scientific interest for WCD. However 1) I'm puzzled by the data. I'm not familiar with ERA5 and E-OBS but reading the data section, it seemed to me that the author actually analyze SWE, not snow depth. It may only be a vocabulary issue.

2) Figure 5 shows that applying a linear regression to annual maxima is not robust since it may be much influenced by 1-2 largest points. Therefore 2 subperiods are considered in Figures 6 to 9, which I support. But then wouldn't it be more consistent to consider in Figures 2-3-4 differences between the two subperiods rather than linear trends? This

is not anecdotal since the regions with largest increase/decrease might partly change (e.g. ITF1). A t-test, e.g., could be applied to test differences in means. Note that another way to get more robust trends in annual maxima is to fit a nonstationary GEV distribution but it may be unnecessarily complicated here.

3) I find the idea of comparing atmospheric fields during extreme events excellent . However I'm puzzled by several interpretations (see below) and I'm not sure that the conclusions are supported by the analysis. First I'd like to see the average Z500 fields for period 2 because I don't think one can interpret anomalies without the mean field (or at least I'm not able to). In Figure 6 the author shows that decreasing trends are mainly associated with negative anomalies over eastern Europe. I see the correlation but is this causality? In particular if one considers a neighboring region with positive trends, don't we have the same pattern (i.e. negative anomalies over EE)? Idem for the positive trends.

4) More generally, looking at the quite noisy map of Figure 4d), is there good hope to be able to explain trends from atmospheric circulation? For example in Italy I can see quite positive, null and negative trends within a few km of a quite flat region. I expect all these regions to be influenced by the same atmospheric circulation, therefore differences in trends are either due to regional characteristics or this is merely rainfall variability (or data issues). Please consider analyzing larger regions to be able interpret smoother maps.

**Specific comments**

L5: "coherent with the mean global warming and previous findings": I'm not sure to understand to which of your results you refer to here.

L6: "discrepancy between trends in average and maximum SD": to investigate this, wouldn't be interesting to look at the regions with the largest discrepancies between means and extremes? Introduction: please consider referring to Beniston et al. 2018, The European mountain cryosphere: a review of its current state, trends, and fu-

ture challenges, which gives a good overview of changes in the European mountain crysophere.

L95: "large SD amounts correspond to snow to be removed": I'm not sure about that. The weight of the snow (SWE) is much more important than the depth.

L 100: "total amount of water": does ER5 really give you a total amount of water? Then this would be a SWE (mm of water), not a depth. Or do you mean "total snow depth"?

L 108: "from daily total precipitation": Idem I don't understand how you get snow depth from water amount.

L113: where does this 2/3" coefficient come from?

Figures 2-3-4; please consider exchanging colors since later on red=decrease, blue=increase.

Please consider merging Figures 2 and 3 (e.g. by crossing out the significant regions)

Figure 4: are you sure these are NUTS-2 regions? It seems to me they are much larger.

L 146-147 "Indeed . . . trends" : actually this was also the case in Fig 3a)

L 164 "due to the two outliers" : I guess these two outliers occurred at the end of the period

Figure 6: please consider showing the average field of period 2. Also the windows are much too large. Please consider showing smaller windows centered on the considered locations.

L 178 "weaker cyclonic structure" : I understand that geopotential heights are higher (positive anomalies) but don't you need the mean field to interpret it as a "weaker cyclonic structure"?

L 179 "an anti-zonal of a blocked pattern": I'm not an expert in atmospheric circulation

but I don't understand where you see that

Figure 8: is the scale the same for all panels?

L 182 "the surrounding . . . events": is tis particular to CZ03? Actually I see that in all panels.

L 187 "negative SD anomalies . . . viceversa": I don't see that (or I don't understand)

L 195 "tend to suggest a stronger meridional flux" : I don't understand this interpretation

L 196 "deeper cyclones" : I don't understand why negative anomalies imply deeper cyclones.

L 220: "we observe more anticyclonic conditions" : where do you show that? I'm not sure that this kind of conclusion can be drawn from a few events.

**Technical corrections:**

L 63: Luthi et al: commas

L 101: "higher" → larger

L 120: "tend coincide"

L 143: "NUTS2" is "NUTS-2" above

L 153: "could hep"

L 155 Altman: commas

Figure 5: NUTs2. Also I guess a) is positive and b) is negative

L167: "atmospheric" → meteo?

L 182 "positive anomalies" → negative?

L 183 "positive SD anomalies" → negative?

L 193: CH5 → CH05
* * *

---

## Author Comment (AC1) · 13 Mar 2020

General comments: In this study, the author investigates recent trends in yearly total snow depth and maximum snow depth in the European region, and for the latter discusses the relationship between the trend and atmospheric circulation and global warming. The reviewer agrees that the manuscript contains a lot of scientific interests to be published since the author focuses on a counterintuitive result: the increasing maximum snow-depth trend under global warming.

**I thank the reviewer for the positive comment. Besides the "counterintuitive result obtained for the Balkan region", I would like to stress that there are also**

[Figure]

**several regions for which snowfall extreme trends are negative. I believe that also for those regions it is important to explain trends in light of the atmospheric circulation, or attribute them to a thermodynamic feedback. I hope that the new version of the manuscript will be clearer for the reviewer and therefore for the readership of WCD.**

The author tried to understand the relationship between the result and change of atmospheric circulation. However, the relationship or causality would be not fully discussed to be published in this manuscript, and in the current status it seems not suitable to the scope of the WCD journal, because the current manuscript contains less investigation on the atmospheric dynamics that causes trends of yearly total and maximum snow-depth. Therefore, I would recommend to resubmit this paper after substantial revision for discussion on the atmospheric dynamics, or the author may address to more elaborate on an observational study such as the comparison with in-situ observations and the ERA5 reanalysis datasets.

**I understand the criticism raised by the referee namely that the manuscript should identify more robust links between the trends and the atmospheric circulation. In the new version of the paper, I include an additional analysis based on the ERA5 dataset decomposition of snowfall into two components: snowfall from large scale precipitation (lsf) and convective snowfall (csf). The use of the two components enable to attribute the changes in the snowfall for the two periods either to large scale flow dynamical changes (and therefore to the anomalies of Z500 fields) or to the thermodynamic changes. Figures 1,2 provide an overview of the climatology of the three components of the snowfall. Following a suggestion of reviewer 2, I have substituted trends with differences between the two subperiods. Indeed, when separating positive and negative trends (see Figure 3), we get that the majority of changes largely due to the large**

**scale snowfall and therefore can be attributed to the extratropical cyclones. The convective snowfall (thermodynamic component) is generally smaller, but convective events can perturb trends in the total snowfall, as their distribution is highly non-Gaussian.**

Specific comments: The conclusions described in Abstract and Conclusions seem not be supported by the results in Sections 2-4. It looks to me only the result that support the conclusion is "This suggests a non-trivial relation between the occurrence of extreme snowfalls, global mean warming and the internal, long-term variability of the atmospheric circulation" (L136-137). Discussions about atmospheric circulations are too few and thus it is difficult to conclude that "the subtle effects of atmospheric circulation in driving extreme events and the non-trivial relation with global warming: a warmer Mediterranean Sea may enhance convective precipitation in winter-time and trigger heavy snowfalls" (L7-9). At least, there is no figures and discussions on specific humidity, climatological temperature that can determine whether snowfall or rainfall, and sea surface temperature and its related surface fluxes (latent and sensible) on Mediterranean Sea.

**The reviewer stresses that there is not enough discussion on the relation between extreme snowfalls and large scale atmospheric circulation VS local convection. I agree that on the first version of the manuscript I have relied on the results of D'Errico et al (2020) to claim that the positive trends on the Mediterranean basin were caused by convective events. I do however agree with the reviewer that more evidence should be provided. For this reason, in the new version of the paper, I will include the analysis on both large scale snowfall and convective snowfall. The new analysis included shows that the trends are largely due to large scale snowfall and not to convective snowfall. The paper will be updated in this direction.**

Also, the reviewer is dissatisfied that the ambiguity of which scale of the atmospheric circulations the author focused on: the synoptic scale or the low-frequency variability? This point was difficult to be understood in Introduction and Section 4.

**I focus on the synoptic scale as opposed to the convective scales. The reviewer would however agree with me that the synoptic scale is not unrelated to the low frequency variability mechanisms: in particular, as shown by D'Errico et al. (2020) there are subseasonal conditions that can favor or not the occurrence of cold and snowy waves over Europe: the presence of stratospheric warming, the presence of an important snow cover on Siberia, the presence of large scale blocking structures. Furthermore, a recent study (Mori et al. 2019 Nature Climate change https://www.nature.com/articles/s41558-018-0379-3) has shown that also the snow cover and the presence of ice in the Arctic region plays an important role. I am aware that this point was not clear in the previous version of the manuscript, in the new one I will be more clear by stating that three different scales (convective, synoptic and sub-seasonal) are relevant for the occurrence of heavy snowfalls and clarify the aspects of the study. In particular, the additional analysis of convective VS large scale snowfall enables to discuss both the aspects.**

Another concern is that the author compared the daily composite fields of the period 1979-1998 with those of 1999-2018 (Figs. 6-9). If my understanding is correct, this comparison is the average of 20 daily fields versus that of 20 fields. It seems to me that the number of composite fields is not enough to discuss the daily atmospheric fields, since the daily fields can emphasize synoptic disturbances such as locations of extratropical cyclones. Thus we may need more larger number of daily fields to be composited, or focus on longer timescale fields for low-frequency variability (e.g., Nakamura et al. 1997). (Nakamura et al. 1997: "The Role of High- and Low-Frequency

C2 Dynamics in Blocking Formation ", Monthly Weather Review)

**The problem of having small samples due to poor quality of snowfall data was already acknowledged in the first version of the manuscript. However, following the suggestion of the reviewer, we can extend the statistics focusing on a 3 day window centered on the event analysed. 3 day seems like a fair choice because this is the typical time scale of development of extratropical cyclones. This allows us to get more robust statistics on the atmospheric circulation features.**

In addition, there would be less discussion on the relationship between atmospheric circulation and global warming. For example, could you compare the increasing/decreasing snow-depth trends with estimation of the Clausius-Clapeyron relationship?

**I believe that the analysis of the convective VS large scale snowfall provided in the paper answers the question raised by the referee.Since no significant trends are present in convective snowfall (see additional figures 1-3) there is no significant thermodynamic effect that could be related to global warming. The trends are largely due to large scale structures.**

Instead, the author could focus on the observational part. I am not familiar with observation research, yet it would be valuable and novel to compare the ERA5 snow estimations with observations. It will provide useful information for reanalyses that are crucially important for weather and climate researches.

**I am afraid this time I have to disagree with the reviewer. I have already used the EOBS dataset (regridded observations) and compared the trends with those**

**provided by ERA5 in the first version of the manuscript.**

It would be helpful to refer to Kawase et al. (2016) who investigated future changes of averaged (yearly total) and extreme (maximum) snowfall events over Japan (East Asian regions), and their results seem partly consistent with your results here. Also you can find Steenburgh and Nakai (2020) for some reviews of snowfall over Japan. (Kawase et al. 2016: Enhancement of heavy daily snowfall in central Japan due to global warming as projected by large ensemble of regional climate simulations, Climatic Change. Steenburgh and Nakai 2020: Perspectives on sea- and lake-effect precipitation from Japan's "Gosetsu Chitai", Bulletin of the American Meteorological Society)

**I thank the reviewer for this inspiring literature, which will be integrated in the new version of the manuscript**

Technical corrections:

-L143: What is the "ERA5 data per NUTS2"? Please describe.

-L153: "hep" => "help".

- What is "the block-maxima procedure"? Please explain.

**All the minor points will be fully addressed in the new version of the manuscript**
**Detailed caption of additional Figures of this review**

- **Figure 1: Climatology of the components of snowfall for the NUTS2 regions. a-b) total snowfall, c-d) snowfall from large scale precipitations (lsf), e-f) snowfall from convective precipitation. a,c,e) average of yearly accumulated snowfall, b,d,f) average of yearly maxima .The data are expressed in units of cm.**

- **Figure 2: Differences of the averages for the (1999-2018) and those for the (1979-1998 ) periods for the NUTS2 regions: a-b) total snowfall, c-d) snowfall from large scale precipitations (lsf), e-f) snowfall from. a,c,e) average of yearly accumulated snowfall, b,d,f) average of yearly maxima .The data are expressed in units of cm.**

- **Figure 3: Convective and large scale snowfall contributions to the differences observed in total snowfall divided by sign of the total snowfall differences: a) positive differences in the mean sf, b) negative differences in the mean sf, c) positive differences in the max sf, d) negative differences in the max sf.**

[Figure]

**Fig. 1.** Climatology of the components of snowfall for the NUTS2 regions.

[Figure]

**Fig. 2.** Differences of the averages for the (1999-2018) and those for the (1979-1998 ) periods for the NUTS2 regions

[Figure]

**Fig. 3.** Convective and large scale snowfall contributions to the differences observed in total snowfall divided by sign of the total snowfall differences

---

## Author Comment (AC2) · 13 Mar 2020

General comments: The article investigates extreme snow depth trends in Europe in the last 40 years and attempts to explain these trends in light of global warming and changes in atmospheric circulation. I find the topic interesting and definitively of scientific interest for WCD.

**I thank the reviewer for the positive comment. In the new version of the manuscript, I will take into full account the comments raised by the reviewer to improve the presentation of the paper.**

[Figure]

1) I'm puzzled by the data. I'm not familiar with ERA5 and E-OBS but reading the data section, it seemed to me that the author actually analyze SWE, not snow depth. It may only be a vocabulary issue.

**Indeed I am using snowfall (sf) data and not snow depth. I have corrected this issue through the paper.**

2) Figure 5 shows that applying a linear regression to annual maxima is not robust since it may be much influenced by 1-2 largest points. Therefore 2 subperiods are considered in Figures 6 to 9, which I support. But then wouldn't it be more consistent to consider in Figures 2-3-4 differences between the two subperiods rather than linear trends? This is not anecdotal since the regions with largest increase/decrease might partly change (e.g. ITF1). A t-test, e.g., could be applied to test differences in means.

**As suggested by the reviewer, the new version of the paper will contain differences between the two sub periods (with a T-test) instead of the linear trends computation. Another important addition will be the use of the two components of snowfall: the large scale snowfall (lsf) and the convective snowfall (cf). They allow to separate the local themodyanmic component of the trends (cf) from the large scale, synoptic component (lsf). Figures 1,2,3 of this review show the results for the three components.**

Note that another way to get more robust trends in annual maxima is to fit a nonstationary GEV distribution but it may be unnecessarily complicated here.

**I have tried the GEV fitting but the results show a very large sensitivity to data. This means that by considering 21 or 19 years, the estimation of the parameters**

**largely fluctuates, whereas it is more stable when considering differences between sub-periods**

3) I find the idea of comparing atmospheric fields during extreme events excellent . However I'm puzzled by several interpretations (see below) and I'm not sure that the conclusions are supported by the analysis. First I'd like to see the average Z500 fields for period 2 because I don't think one can interpret anomalies without the mean field (or at least I'm not able to). In Figure 6 the author shows that decreasing trends are mainly associated with negative anomalies over eastern Europe. I see the correlation but is this causality? In particular if one considers a neighboring region with positive trends, don't we have the same pattern (i.e. negative anomalies over EE)? Idem for the positive trends.

**Thank you for the encouraging comment. In order to progress in the interpretation of the results, in the new version of the manuscript I have used both the large scale and convective snowfall components of the ERA5 dataset. The use of the two components enable to attribute the changes in the snowfalls in the two periods either to large scale flow dynamical changes (and therefore to the anomalies of Z500 fields) or to the thermodynamic changes. Figures 1-3 at the end of this review answer will be added to the new version of the paper. They show that the contribution to the negative or positive trends come mostly from large scale snowfall components. I will therefore simplify the discussion on the circulation patterns in the new version of the paper.**

4) More generally, looking at the quite noisy map of Figure 4d), is there good hope to be able to explain trends from atmospheric circulation? For example in Italy I can see quite positive, null and negative trends within a few km of a quite flat region. I expect all these regions to be influenced by the same atmospheric circulation, therefore

differences in trends are either due to regional characteristics or this is merely rainfall variability (or data issues). Please consider analyzing larger regions to be able interpret smoother maps.

**In the new version of the manuscript, trends will be separated by convective snowfall (local and thermodynamic) and large scale snowfall (regional and dynamical). This analysis provides a more precise assessment of the origin of the trends. The suggestion of looking at larger regions has also been attempted but discarded (NUTS0) because the regions are too large to provide a coherent picture of the large scale atmospheric circulation. Let us consider the emblematic case of Italt: the Italian peninsula is crossed by two mountains ranges (Alps and Apennines), plains. Large countries include geographical features that can indeed trigger a large variability in the snowfall in neighboring regions . Indeed, the suggestion of considering differences in the quantity for the two sub periods instead of linear trends, improve the understanding of the modifications.**

Specific comments

L5: "coherent with the mean global warming and previous findings": I'm not sure to understand to which of your results you refer to here.

**I will rephrase this sentence as: "coherent with previous studies: they allow to link the decrease in snowfall to mean global warming"**

L6: "discrepancy between trends in average and maximum SD": to investigate this, wouldn't be interesting to look at the regions with the largest discrepancies between means and extremes? Introduction: please consider referring to Beniston et al.

2018, The European mountain cryosphere: a review of its current state, trends, and future challenges, which gives a good overview of changes in the European mountain crysophere.

**Thank you for the additional reference that will be added to the new version of the paper.**

L95: "large SD amounts correspond to snow to be removed": I'm not sure about that. The weight of the snow (SWE) is much more important than the depth.

**As answered in comment 1, there was a problem in naming the variable used in the first version of the manuscript. SD will be changed to "snowfall" and references to SD will be dropped.**

L 100: "total amount of water": does ER5 really give you a total amount of water? Then this would be a SWE (mm of water), not a depth. Or do you mean "total snow depth"?

**Again, thanks for pointing out the problems with this variable. In the new version of the manuscript it will be changed to snowfall.**

L 108: "from daily total precipitation": Idem I don't understand how you get snow depth from water amount.

**Also here I will change the description to point out that from total daily precipitation we can get a proxy of snowfall (and not snow depth) variables.**

L113: where does this 2/3" coefficient come from? Figures 2-3-4; please consider exchanging colors since later on red=decrease, blue=increase. Please consider merging Figures 2 and 3 (e.g. by crossing out the significant regions)

**The coefficient $\frac{2}{3}$ is the best match between the EOBS and the ERA5 data, this will be added to the text. I will merge figure 2 and 3 as suggested. I prefer to keep red and blue in Figure 2 and 3 and exchange them in the boxplot figures for coherence.**

Figure 4: are you sure these are NUTS-2 regions? It seems to me they are much larger. L 146-147 "Indeed . . . trends" : actually this was also the case in Fig 3a)

**According to wikipedia, the NUTS-2 regions used in the paper are correct: https://en.wikipedia.org/wiki/Nomenclature_of_Territorial_Units_for_Statistics The level of "department" or "province" is NUTS-3 while the level of states is NUTS-1**

L 164 "due to the two outliers" : I guess these two outliers occurred at the end of the period
**That is right, I will specify this in the text of the next version of the manuscript.**

Figure 6: please consider showing the average field of period 2. Also the windows are much too large. Please consider showing smaller windows centered on the considered locations.

**I will follow the suggestions by the reviewer in the next version of the manuscript.**

L 178 "weaker cyclonic structure" : I understand that geopotential heights are higher (positive anomalies) but don't you need the mean field to interpret it as a "weaker cyclonic structure"? L 179 "an anti-zonal of a blocked pattern": I'm not an expert in atmospheric circulation but I don't understand where you see that Figure 8: is the scale the same for all panels? L 182 "the surrounding . . . events": is tis particular to CZ03? Actually I see that in all panels. L 187 "negative SD anomalies . . . viceversa": I don't see that (or I don't understand) L 195 "tend to suggest a stronger meridional flux" : I don't understand this interpretation L 196 "deeper cyclones" : I don't understand why negative anomalies imply deeper cyclones. L 220: "we observe more anticyclonic conditions" : where do you show that? I'm not sure that this kind of conclusion can be drawn from a few events.

**I will address all the comments relative to the discussion of synoptic trends in the manuscript focusing more on the differences between csf and lsf and their contribution to the total sf trends.**

Technical corrections: L 63: Luthi et al: commas L 101: "higher" → larger L 120: "tend coincide" L 143: "NUTS2" is "NUTS-2" above L 153: "could hep" L 155 Altman: commas Figure 5: NUTs2. Also I guess a) is positive and b) is negative L167: "atmospheric" → meteo? L 182 "positive anomalies" → negative? L 183 "positive SD anomalies" → negative? L 193: CH5 → CH05

**All the minor points will be fully addressed in the new version of the manuscript**

**Detailed caption of additional Figures of this review**

- **Figure 1: Climatology of the components of snowfall for the NUTS2 regions. a-b) total snowfall, c-d) snowfall from large scale precipitations**

(lsf), e-f) snowfall from convective precipitation. a,c,e) average of yearly accumulated snowfall, b,d,f) average of yearly maxima .The data are expressed in units of cm.

- **Figure 2: Differences of the averages for the (1999-2018) and those for the (1979-1998 ) periods for the NUTS2 regions: a-b) total snowfall, c-d) snowfall from large scale precipitations (lsf), e-f) snowfall from. a,c,e) average of yearly accumulated snowfall, b,d,f) average of yearly maxima .The data are expressed in units of cm.**

- **Figure 3: Convective and large scale snowfall contributions to the differences observed in total snowfall divided by sign of the total snowfall differences: a) positive differences in the mean sf, b) negative differences in the mean sf, c) positive differences in the max sf, d) negative differences in the max sf.**

[Figure]

[Figure]

**Fig. 1.** Climatology of the components of snowfall for the NUTS2 regions.

[Figure]

**Fig. 2.** Differences of the averages for the (1999-2018) and those for the (1979-1998 ) periods for the NUTS2 regions

[Figure]

**Fig. 3.** Convective and large scale snowfall contributions to the differences observed in total snowfall divided by sign of the total snowfall differences

---

## Editor Comment (EC1) · Christian M. Grams (Editor) · 19 Mar 2020

Dear Davide Faranda,

as you have seen we have now received two thorough reviews for your manuscript WCD-2019-15 "An attempt to explain recent trends in European snowfall extremes". Thank you for your initial replies to the reviewer comments and thoughts on how to address their major concerns in a revised version.

While I agree that your study and the documentation of snowfall trends is an important topic and in the scope of WCD, I concur with both reviewers, that the dynamical interpretation is not elaborate enough in the current version to explain the observed trend. Having also consulted the editors of the journal this is a major issue which must be

addressed prior to publication in WCD.

Therefore a more thorough investigation of the drivers for the observed trends is needed. The reviewers give valuable input in this direction. In particular, the example of how ambiguous trends in Japanese snowfall can be explained by the the JPCZ in Kawase et al. 2016 (Section 4) is an excellent example of a potential dynamical interpretation in their case. Evidence in that direction should be presented; although I see the difficulty in finding common drivers for the various focus regions you investigated compared to them.

We are afraid that your suggestion to investigate convective vs. large-scale snowfall from ERA-5 will not solve the problem. This separation highly depends on the grid spacing in the model and can therefore not be a proper indicator of a physical separation into thermodynamic and dynamic processes. Convective snowfall simply shows the fraction which is processed in the physical schemes vs. dynamical core (or resolved vs. unresolved scales) and therefore not necessarily attributable to actual convection.

It is striking that most of your positive trend locations are along Mediterranean coasts and it is likely that the large-scale circulation is involved. I would therefore suggest that you directly show the synoptic environment during these snowfall events. E.g. parameters indicating stability, flow direction towards the coast and local orography. This would yield a more dynamical interpretation, e.g. that starker sea surface - troposphere temperature contrasts might enhance moisture uptake combined with reduced stability aloft, change ascent, local stability, and local convergence during snowfall events. The body of literature looking at cold-air outbreaks, air-sea interaction, and baroclinicity might give additional guidance here (e.g. Czaja et al. 2019, Papritz and Spengler 2015, 2016). The role of blocking remains obscure although the abstract suggests you would look into this dynamical aspect in more detail (I see that this is ambiguous terminology with "block-maxima" referring to snow depth, but WCD readers might relate this to blocking anticyclones). Apart from this Eulerian investigation a Lagrangian investigation of air mass origin (location and changes of physical properties along trajectories) contrasting events in the early and late decades would also give dynamical insight. Another thought is that these trends might be driven by decadal variability, an aspect that could be discussed more comprehensively (the 2000s were more NAO-compared to the 1990s, e.g. Weisheimer et al. 2017).

Finally I outline a thought that has arisen from the discussion amongst the editors. The thermodynamic argument that a warmer Mediterranean Sea may lead to larger snowfall amounts at some places in Europe requires caution and might be flawed. The reason for this is that snowfall extremes (at least in the midlatitudes) tend to occur at or near the freezing point in both colder and warmer climates (see e.g. O'Gorman et al. 2014). Also the results in the present paper indicate that locally (at the location of the snowfall) the temperature difference between the cold and warm period is small (Fig. 7, although this is a bit hard to see). Now, it is this local temperature that determines the maximum atmospheric moisture content and thus the "thermodynamic" component of the snowfall amount. If the Mediterranean Sea warms a lot, enhancing evaporation there, but the temperature at the location of the snowfall stays the same, then all the excess moisture precipitates out during the transport to this location, and there is no substantial "thermodynamic" enhancement of the snowfall. So the question arises if there are more snowfall events rather than more intense events and/or if dynamics enhance the events rather than thermodynamics, again pointing towards variability of the large-scale circulation between the considered periods....

These are some thoughts, but I hope this gives direction on how more dynamical evidence can be given in order to address this critical point.

Kind regards

Christian M. Grams

Czaja, A., C. Frankignoul, S. Minobe, and B. Vannière, 2019: Simulating the Midlatitude Atmospheric Circulation: What Might We Gain From High-Resolution Modeling of Air-Sea Interactions? Curr Clim Change Rep, doi:10.1007/s40641-019-00148-5.

O'Gorman, P. A., 2014: Contrasting responses of mean and extreme snowfall to climate change. Nature, 512, 416–418, doi:10.1038/nature13625.

Papritz, L., and T. Spengler, 2016: A Lagrangian Climatology of Wintertime Cold Air Outbreaks in the Irminger and Nordic Seas and Their Role in Shaping Air–Sea Heat Fluxes. J. Climate, 30, 2717–2737, doi:10.1175/JCLI-D-16-0605.1.

Papritz, L., and T. Spengler, 2015: Analysis of the slope of isentropic surfaces and its tendencies over the North Atlantic. Q.J.R. Meteorol. Soc., 141, 3226–3238, doi:10.1002/qj.2605.

Weisheimer, A., N. Schaller, C. O'Reilly, D. A. MacLeod, and T. Palmer, 2017: Atmospheric seasonal forecasts of the twentieth century: multi-decadal variability in predictive skill of the winter North Atlantic Oscillation (NAO) and their potential value for extreme event attribution. Q.J.R. Meteorol. Soc., 143, 917–926, doi:10.1002/qj.2976.
* * *

---

## Author Response (AR1)

**Dear Editor,**

**I have revised the article following the editorial suggestions provided and I have therefore greatly modified the article as well as the answers provided to the reviewers. In particular, as suggested by the editorial board of WCD, I have dropped the analysis of large scale versus convective snowfall. This deep revision of the article is articulated on the following different axes:**

1) **More tests to check consistency between EOBS and ERA5 datasets**
2) **The analysis of trends at the level of NUTS0 (countries) as suggested by one of the reviewer**
3) **A more careful analysis of thermodynamic and dynamic factors that can explain positive trends, on the line of editorial requests. For the thermodynamics I now use CAPE and t2m fields. For the dynamics I perform a sea-level-pressure analysis as well as a weather regimes analysis and an analogs search.**
4) **Extension of the bibliography to support the claims made in the manuscript.**

**I provide below a detailed answer to both editorial and referees comments and I invite the referees to base their future comments on the answers provided hereby and not on those posted on the interactive discussion. I hope that this new version of the manuscript will be suitable for publication in WCD.**

**Best Regards,**
**Davide Faranda**

**ANSWERS TO EDITORIAL COMMENTS**

While I agree that your study and the documentation of snowfall trends is an important topic and in the scope of WCD, I concur with both reviewers, that the dynamical interpretation is not elaborate enough in the current version to explain the observed trend. Having also consulted the editors of the journal this is a major issue which must be addressed prior to publication in WCD. Therefore a more thorough investigation of the drivers for the observed trends is needed.

**I wish to acknowledge the editorial board for finding the time to provide an additional review of the paper and I have undertaken the steps suggested by the board to improve the manuscript.**

The reviewers give valuable input in this direction. In particular, the example of how ambiguous trends in Japanese snowfall can be explained by the the JPCZ in Kawase et al. 2016 (Section 4) is an excellent example of a potential dynamical interpretation in their case. Evidence in that direction should be presented; although I see the difficulty in finding common drivers for the various focus regions you investigated compared to them.

**Following the suggestion of the editorial board, I have performed both a weather regime analysis and analogs search of the atmospheric circulation patterns associated with heavy snowfalls. The detection of cyclonic and anticyclonic structures is now based on the sea-level pressure, instead of the geopotential height. Absolute and anomalies fields are commented, as suggested by reviewer 2. The dynamical analysis suggests that there is a prevalence of Atlantic Ridge patterns associated with the extreme events with a tendency to increase in recent times. The new section 4.2 (and Figures 9-11) is entirely dedicated to the connection between heavy snowfalls and weather regimes.**

We are afraid that your suggestion to investigate convective vs. large-scale snowfall from ERA-5 will not solve the problem. This separation highly depends on the grid spacing in the model and can therefore not be a proper indicator of a physical separation into thermodynamic and dynamic processes. Convective snowfall simply shows the fraction which is processed in the physical schemes vs. dynamical core (or resolved vs. unresolved scales) and therefore not necessarily attributable to actual convection.

**Indeed, I agree with the editorial board on this point. I have also consulted our modellists and they agree that the separation is somehow artificial and not attributable to actual convection. In the new version of the manuscript the role of convection has been assessed using convective available potential energy (CAPE).**

It is striking that most of your positive trend locations are along Mediterranean coasts and it is likely that the large-scale circulation is involved. I would therefore suggest that you directly show the synoptic environment during these snowfall events. E.g. parameters indicating stability, flow direction towards the coast and local orography. This would yield a more dynamical interpretation, e.g. that starker sea surface - troposphere temperature contrasts might enhance moisture uptake combined with reduced stability aloft, change ascent, local stability, and local convergence during snowfall events. The body of literature looking at cold-air outbreaks, air-sea interaction, and baroclinicity might give additional guidance here (e.g. Czaja et al. 2019, Papritz and Spengler 2015, 2016).

**As suggested both by the editor and by reviewer #2, the new version of the manuscript focuses on positive trends and at the scales of countries. I have identified four countries with positive trends to perform the analyses: Albania, Macedonia, Switzerland and Turkey. Instead of looking at convective snowfall, the role of convection is now analysed through the CAPE as suggested in the existing literature. CAPE analysis shows that there is indeed an increase of convective instability for the second period and the high CAPE values are presented over the Mediterranean Sea, thus I cannot exclude that thermodynamic factor plays along the dynamic factors to enhance snowfalls.**

The role of blocking remains obscure although the abstract suggests you would look into this dynamical aspect in more detail (I see that this is ambiguous terminology with "block-maxima" referring to snow depth, but WCD readers might relate this to blocking anticyclones). Apart from this Eulerian investigation a Lagrangian investigation of air mass origin (location and changes of physical properties along trajectories) contrasting events in the early and late decades would also give dynamical insight. Another thought is that these trends might be driven by decadal variability, an aspect that could be discussed more comprehensively (the 2000s were more NAOcompared to the 1990s, e.g. Weisheimer et al. 2017).

**Following the suggestion, I have better investigated the role of meridional patterns in the variability of extreme snowfalls. For the four countries analysed, there is a tendency to an increase of Atlantic Ridge patterns during extreme events, although the tendency in the analogs of those days is weak and non-significant. I hope that the new analysis in Figures 9-11 is more convincing and supporting than the one presented in the previous version of the manuscript.**

Finally I outline a thought that has arisen from the discussion amongst the editors. The thermodynamic argument that a warmer Mediterranean Sea may lead to larger snowfall amounts at some places in Europe requires caution and might be flawed. The reason for this is that snowfall extremes (at least in the midlatitudes) tend to occur at or near the freezing point in both colder and warmer climates (see e.g. O'Gorman et al. 2014). Also the results in the present paper indicate that locally (at the location of the snowfall) the temperature difference between the cold and warm period is small (Fig. 7, although this is a bit hard to see). Now, it is this local temperature that determines the maximum atmospheric moisture content and thus the "thermodynamic" component of the snowfall amount. If the Mediterranean Sea warms a lot, enhancing evaporation there, but the temperature at the location of the snowfall stays the same, then all the excess moisture precipitates out during the transport to this location, and there is no substantial "thermodynamic" enhancement of the snowfall. So the question arises if there are more snowfall events rather than more intense events and/or if dynamics enhance the events rather than thermodynamics, again pointing towards variability of the large-scale circulation between the considered periods.

**The arguments raised by the editorial board about the role of the thermodynamic vs dynamical components of changes are interesting and I have tried to address them in the new version of the manuscript. I do agree that atmospheric circulation plays an important role in the trends, as shown by the prevalence of Atlantic Ridge patterns evidenced by the analyses in Figure 10, but I am still convinced that there is an interplay of circulation and thermodynamic factors to explain the observed trends: the analysis of CAPE shows that large values of this quantity are associated with heavy snowfalls in the selected countries. CAPE values of 70 JKg-1 are enough to trigger convection during winter time and enhance snowfall precipitations (Olsson et al. 2017). For all countries analysed, the isobars associated with the cyclonic conditions embedded in Atlantic ridge patterns point to winds blowing from sea to land, thus favoring the advection of moisture and the**

**formation of convective precipitation. In addition, the four countries analysed are characterised by mountain ranges that, in presence of sea-to-land flow, favors Stau effects. Both thermodynamics and dynamics effects seem therefore to contribute to observed trends, although it is difficult to understand which factor prevails.**

**ANSWERS TO REVIEWER 1**

Comments on " An attempt to explain recent trends in European snowfall extremes" by Dr. Davide Faranda submitted to WCD.

General comments: In this study, the author investigates recent trends in yearly total snow depth and maximum snow depth in the European region, and for the latter discusses the relationship between the trend and atmospheric circulation and global warming. The reviewer agrees that the manuscript contains a lot of scientific interests to be published since the author focuses on a counterintuitive result: the increasing maximum snow-depth trend under global warming.

**I thank the reviewer for the positive comment. In the new version of the manuscript have provided more insights on the thermodynamic and dynamical factors that could cause such trends**

The author tried to understand the relationship between the result and change of atmospheric circulation. However, the relationship or causality would be not fully discussed to be published in this manuscript, and in the current status it seems not suitable to the scope of the WCD journal, because the current manuscript contains less investigation on the atmospheric dynamics that causes trends of yearly total and maximum snow-depth. Therefore, I would recommend to resubmit this paper after substantial revision for discussion on the atmospheric dynamics, or the author may address to more elaborate on an observational study such as the comparison with in-situ observations and the ERA5 reanalysis datasets.

**I understand the criticism raised by the referee namely that the manuscript should identify more robust links between the trends and the atmospheric circulation. In the new version of the paper, I have performed both a weather regime analysis and analogs search of the atmospheric circulation patterns associated with heavy snowfalls. This analysis is useful to show that there is a prevalence of Atlantic Ridge patterns associated with the extreme events with a tendency to increase in recent times. The new section 4.2 (and Figures 9-11) is entirely dedicated to the connection between heavy snowfalls and weather regimes.**

Specific comments: The conclusions described in Abstract and Conclusions seem not be supported by the results in Sections 2-4. It looks to me only the result that support the conclusion is "This suggests a non-trivial relation between the occurrence of extreme snowfalls,

global mean warming and the internal, long-term variability of the atmospheric circulation" (L136-137). Discussions about atmospheric circulations are too few and thus it is difficult to conclude that "the subtle effects of atmospheric circulation in driving extreme events and the non-trivial relation with global warming: a warmer Mediterranean Sea may enhance convective precipitation in winter-time and trigger heavy snowfalls" (L7-9). At least, there is no figures and discussions on specific humidity, climatological temperature that can determine whether snowfall or rainfall, and sea surface temperature and its related surface fluxes (latent and sensible) on Mediterranean Sea.

**The reviewer stresses that there is not enough discussion on the relation between extreme snowfalls and large scale atmospheric circulation VS local convection. I agree that in the first version of the manuscript I have relied on the results of D'Errico et al (2020) to claim that the positive trends on the Mediterranean basin were caused by convective events. I do however agree with the reviewer that more evidence should be provided. For this reason, in the new version of the paper, I include the analysis of the stability via the convective available potential energy. This analysis shows indeed that, for the countries showing positive trends, the instability increases in the second period, thus supporting the claim that thermodynamics play a role in snowfall changes.**

Also, the reviewer is dissatisfied that the ambiguity of which scale of the atmospheric circulations the author focused on: the synoptic scale or the low-frequency variability? This point was difficult to be understood in Introduction and Section 4.

**In the new version of the manuscript, it is made clear that this study focuses on synoptic scales. In the introduction I have added (L30-L32): "The goal of this paper is to shed a light on recent changes in the dynamics of extreme snowfalls, by projecting the recent changes in frequency/intensity of extreme snowfalls on the large scale (synoptic) dynamical drivers and identifying possible small scale convective thermodynamic feedback." The role of low frequency variability is discussed in the conclusions (L300-306): "There are sub-seasonal to seasonal conditions that can trigger snowy waves over Europe by modifying winter atmospheric circulation patterns: the role of stratospheric warming, the magnitude of snow cover on Siberia and in the Arctic region could be taken into account in future research on this topic, e.g. by following the approaches of ~\citet{handorf2015impacts,handorf2017arctic} and \citet{mori2019reconciled}. At smaller scales, where convection is important, further studies could be based on searching the origin, transport pathways, and thermodynamic evolution of air masses involved in heavy snowfall episodes, via novel methodologies based on tracking trajectories of air masses as those introduced in ~\citet{papritz2017lagrangian}, and by using convection permitting models to study sea-air-snow interactions~\citep{bartolini2019convection}."**

Another concern is that the author compared the daily composite fields of the period 1979-1998 with those of 1999-2018 (Figs. 6-9). If my understanding is correct, this comparison is the average of 20 daily fields versus that of 20 fields. It seems to me that the number of composite fields is not enough to discuss the daily atmospheric fields, since the daily fields can emphasize synoptic disturbances such as locations of extratropical cyclones. Thus we may need more larger number of daily fields to be composited, or focus on longer timescale fields for low-frequency variability (e.g., Nakamura et al. 1997). (Nakamura et al. 1997: "The Role of High- and Low-Frequency C2 Dynamics in Blocking Formation ", Monthly Weather Review)

**The problem of having small samples due to poor quality of snowfall data was already acknowledged in the first version of the manuscript. However, following the suggestion of the reviewer, I have performed an analogs analysis, to detect trends in the patterns associated with the occurrence of heavy snowfalls (typically Atlantic ridge). As analogs, I select the 5% closest patterns to the average sea-level pressure patterns identified, for the two periods, during snow events. Note that the results do not depend on the threshold used for the selection in the range 0.25% to 5%. This analysis shows that, besides a significant time decreasing trend for analogs of the typical situations leading to snowfall in Switzerland, there are no long term trends in analogs for the other countries examined.**

In addition, there would be less discussion on the relationship between atmospheric circulation and global warming. For example, could you compare the increasing/decreasing snow-depth trends with estimation of the Clausius-Clapeyron relationship?

**I have better explored the possibility of using the Clausius Clapeyron relationship but I have not found, in the literature, a common agreement on the possibility of applying it locally in space and time, as it would be needed to study snowfall extremes. For example, PA O'Gorman, CJ Muller - Environmental Research Letters, 2010 find deviations from the relation that can be both attributed to physics or to model capabilities. I would avoid going into this slippery argument. However, if the reviewer has a convincing argument and a practical suggestion to undertake this analysis, I would be happy to change my mind.**

Instead, the author could focus on the observational part. I am not familiar with observation research, yet it would be valuable and novel to compare the ERA5 snow estimations with observations. It will provide useful information for reanalyses that are crucially important for weather and climate researches.

**I am afraid but this time I have to disagree with the reviewer. I have already used the EOBS dataset (regridded observations) and compared the trends with those provided by ERA5 in the first version of the manuscript.**

It would be helpful to refer to Kawase et al. (2016) who investigated future changes of averaged (yearly total) and extreme (maximum) snowfall events over Japan (East Asian regions), and their results seem partly consistent with your results here. Also you can find Steenburgh and Nakai (2020) for some reviews of snowfall over Japan. (Kawase et al. 2016: Enhancement of heavy daily snowfall in central Japan due to global warming as projected by large ensemble of regional climate simulations, Climatic Change. Steenburgh and Nakai 2020: Perspectives on sea- and lake-effect precipitation from Japan's "Gosetsu Chitai", Bulletin of the American Meteorological Society)

**I thank the reviewer for this inspiring literature, which will be integrated in the new version of the manuscript. Following the approach of Kawase et al. 2016, I have performed both a weather regime analysis and analogs search of the atmospheric circulation patterns associated with heavy snowfalls. This analysis is useful to show that there is a prevalence of Atlantic Ridge patterns associated with extreme snowfall events.**

Technical corrections:

-L143: What is the "ERA5 data per NUTS2"? Please describe.

**I have added: "We show results at two different levels, regional (NUTS2) and national (NUSTS0). These subdivisions are commonly used by stake-holders to assess impacts of climate variables on economy and society and are the reference adopted by Copernicus for its products (see, e.g.~\citep{brandmueller2017territorial})"**

-L153: "hep" => "help".

**corrected**

- What is "the block-maxima procedure"? Please explain.

**The reference to block maxima has been dropped as it was confusing.**

**ANSWERS TO REVIEWER 2**

**General comments:**
The article investigates extreme snow depth trends in Europe in the last 40 years and attempts to explain these trends in light of global warming and changes in atmospheric circulation. I find the topic interesting and definitively of scientific interest for WCD.

**I thank the reviewer for the positive comment. In the new version of the manuscript, I will take into full account the comments raised by the reviewer to improve the presentation of the paper.**

1) I'm puzzled by the data. I'm not familiar with ERA5 and E-OBS but reading the data section, it seemed to me that the author actually analyze SWE, not snow depth. It may only be a vocabulary issue.

**Indeed I am using snowfall (sf) data and not snow depth. I have corrected this issue through the paper.**

2) Figure 5 shows that applying a linear regression to annual maxima is not robust since it may be much influenced by 1-2 largest points. Therefore 2 subperiods are considered in Figures 6 to 9, which I support. But then wouldn't it be more consistent to consider in Figures 2-3-4 differences between the two subperiods rather than linear trends? This is not anecdotal since the regions with largest increase/decrease might partly change (e.g. ITF1). A t-test, e.g., could be applied to test differences in means.

**As suggested by the reviewer, the new version of the paper contains differences between the two sub periods (with a T-test for significance) instead of the linear trends computation.**

Note that another way to get more robust trends in annual maxima is to fit a nonstationary GEV distribution but it may be unnecessarily complicated here.

**I have tried the GEV fitting but the results show a very large sensitivity to data. This means that by considering 21 or 19 years, the estimation of the parameters largely fluctuates, whereas it is more stable when considering differences between sub-periods**

3) I find the idea of comparing atmospheric fields during extreme events excellent . However I'm puzzled by several interpretations (see below) and I'm not sure that the conclusions are supported by the analysis. First I'd like to see the average Z500 fields for period 2 because I don't think one can interpret anomalies without the mean field (or at least I'm not able to). In Figure 6 the author shows that decreasing trends are mainly associated with negative anomalies over eastern Europe. I see the correlation but is this causality? In particular if one considers a neighboring region with positive trends, don't we have the same pattern (i.e. negative anomalies over EE)? Idem for the positive trends.

**Thank you for the encouraging comment. In order to progress in the interpretation of the results, in the new version of the manuscript I have largely extended the analysis. First of all, I moved from Z500 to the sea-level pressure fields to track the circulation. This choice is motivated by the fact that Z500 presents a thermodynamic trend superimposed to the information about the circulation (Jézéquel, Aglaé, et al. "Trends of atmospheric**

circulation during singular hot days in Europe." *Environmental Research Letters* **13.5 (2018): 054007.). As suggested I now plot both the absolute fields as well as the anomalies, so the patterns are more recognizable and directions of the atmospheric flow can be justified. This analysis (L227-240) allows us to identify cyclonic structures such as the Genoa Low or the Cyprus Low associated with extreme snowfalls. The anomalies show an enhancement of those patterns, possibly enhancing extreme snowfalls.**

4) More generally, looking at the quite noisy map of Figure 4d), is there good hope to be able to explain trends from atmospheric circulation? For example in Italy I can see quite positive, null and negative trends within a few km of a quite flat region. I expect all these regions to be influenced by the same atmospheric circulation, therefore differences in trends are either due to regional characteristics or this is merely rainfall variability (or data issues). Please consider analyzing larger regions to be able interpret smoother maps.

**Following the suggestion of the reviewer, in the new version of the paper, I also use the NUTS0 level (country) to investigate trends. NUTS-0 is the scale of European countries. This scale allows us to identify that the increase in maximum snowfall is a stable feature in the Balkans. Four countries showing positive trends are then analysed in more detail than in the previous version, focusing on both thermodynamics and dynamics aspects. For thermodynamics, I analyse atmospheric stability via convective available potential energy (CAPE). This analysis shows an increase of instability in the second period with values that, in previous studies (Olsson et al. 2017) have been identified as sufficient to trigger snowfall convective precipitations. Furthermore, I perform both a weather regime analysis and analogs search of the atmospheric circulation patterns associated with heavy snowfalls. This analysis is useful to show that there is a prevalence of Atlantic Ridge patterns associated with the extreme events with a tendency to increase in recent times. The new section 4.2 (and Figures 9-11) is entirely dedicated to the connection between heavy snowfalls and weather regimes. The combination of these analyses show that there is an interplay between dynamical and thermodynamic factors in determining the changes in snowfall maxima.**

**Specific comments**

L5: "coherent with the mean global warming and previous findings": I'm not sure to understand to which of your results you refer to here.

**I will rephrase this sentence as: "is coherent with previous findings and caused by global warming"**

L6: "discrepancy between trends in average and maximum SD": to investigate this, wouldn't be interesting to look at the regions with the largest discrepancies between means and extremes? Introduction: please consider referring to Beniston et al. 2018, The European mountain cryosphere: a review of its current state, trends, and future challenges, which gives a good overview of changes in the European mountain crysophere.

**Thank you for the suggestion. Indeed I analyse, on one hand, two countries in the Balkans because they have positive or zero trends also for total yearly SF and, on the other hand, Switzerland and Turkey because they show opposite trends between mean and extremes (the largest discrepancy). Thank you for the additional reference that will be added to the new version of the paper.**

L95: "large SD amounts correspond to snow to be removed": I'm not sure about that. The weight of the snow (SWE) is much more important than the depth.

**As answered in comment 1, there was a problem in naming the variable used in the first version of the manuscript. SD will be changed to "snowfall" and references to SD will be dropped.**

L 100: "total amount of water": does ER5 really give you a total amount of water? Then this would be a SWE (mm of water), not a depth. Or do you mean "total snow depth"?

**Again, thanks for pointing out the problems with this variable. In the new version of the manuscript it will be changed to snowfall.**

L 108: "from daily total precipitation": Idem I don't understand how you get snow depth from water amount.

**Also here I will change the description to point out that from total daily precipitation we can get a proxy of snowfall (and not snow depth) variables.**

L113: where does this 2/3" coefficient come from? Figures 2-3-4; please consider exchanging colors since later on red=decrease, blue=increase. Please consider merging Figures 2 and 3 (e.g. by crossing out the significant regions)

**The coefficient 2/3 has been dropped in the new version of the paper, which also uses the newest EOBS dataset. I prefer to keep red and blue in Figure 2 and 3 .**

Figure 4: are you sure these are NUTS-2 regions? It seems to me they are much larger. L 146-147 "Indeed . . . trends" : actually this was also the case in Fig 3a)

**According to wikipedia, the NUTS-2 regions used in the paper are correct:**
**https://en.wikipedia.org/wiki/Nomenclature_of_Territorial_Units_for_Statistics**

**The level of "department" or "province" is NUTS-3 while the level of states is NUTS-0. NUTS are now better referenced.**

L 164 "due to the two outliers" : I guess these two outliers occurred at the end of the period

**That is right and it is specified this in the text of the next version**

Figure 6: please consider showing the average field of period 2. Also the windows are much too large. Please consider showing smaller windows centered on the considered locations.

**I have followed the suggestion of the reviewer and show absolute fields, together with the anomalies. Focus on countries, instead of regions, improves the visualization of changes. A more focused window on the interested countries has been selected.**

L 178 "weaker cyclonic structure" : I understand that geopotential heights are higher (positive anomalies) but don't you need the mean field to interpret it as a "weaker cyclonic structure"?  L 179 "an anti-zonal of a blocked pattern": I'm not an expert in atmospheric circulation but I don't understand where you see that Figure 8: is the scale the same for all panels? L 182 "the surrounding . . . events": is tis particular to CZ03? Actually I see that in all panels. L 187 "negative SD anomalies . . . viceversa": I don't see that (or I don't understand)  L 195 "tend to suggest a stronger meridional flux" : I don't understand this interpretation  L 196 "deeper cyclones" : I don't understand why negative anomalies imply deeper cyclones. L 220: "we observe more anticyclonic conditions" : where do you show that? I'm not sure that this kind of conclusion can be drawn from a few events.

**All this part will be completely rewritten. First of all geopotential height has been substituted by sea-level pressure, which allows to visualize better cyclonic structures (that are indeed present). All of these comments are therefore answered in the new version of the paper**

 ###Technical corrections:
L 63: Luthi et al: commas
L 101: "higher" → larger
L 120: "tend coincide"
L 143: "NUTS2" is "NUTS-2" above
L 153: "could hep"
L 155 Altman: commas
Figure 5: NUTs2. Also I guess a) is positive and b) is negative
L167: "atmospheric" → meteo? L 182 "positive anomalies" → negative?
L 183 "positive SD anomalies" → negative?
L 193: CH5 → CH05

**All the technical corrections will be taken into account**

---

## Referee Report (RR1)

Final review comment:

Thank you again for your much effort to revise the manuscript. I
have no additional comments before acceptance except to remove
Nakamura et al. (1997) from L45.

---

## Author Response (AR2)

**Answers to Editorial Comments**

I thank the author for a major revision of the manuscript. The reviewers and I appreciate your effort in substantially improving the manuscript. However there are still some critical issues that require major revision before the paper could be considered for final publication in WCD.

**Thank you for the encouraging comments. I definitely appreciate the time and the effort the reviewers and the editorial board have put in considering my work. I have taken into account the remarks and I am happy to provide a new version of the manuscript which addresses the issues.**

For instance, reviewer 2 points towards critical discrepancies between the snow rates in the EOBS data and ERA-5 which lead to statements not justified by both data sets and contradict findings related to snow-fall trends in CH in earlier literature. A work around could be to focus on the consistent trends in both data sets e.g. in maximum snowfall at the Adriatic Coast as suggested by R2.

**I agree that there are critical discrepancies between EOBS and ERA5 data and this leads to contradicting findings for Switzerland. In the new version of the manuscript, I discuss this contradiction briefly and I focus only on consistent trends for both datasets for countries located in the Balkans as suggested by reviewer 2.**

Another issue raised by reviewer 1 is the need for a clearer distinction of the different time scales involved or at least a clearer introduction to different time scales is needed. I hope that these second reviews encourage you for another major revision.

**The time scales are now discussed in detail in the introduction (L35-40): "The focus of this study is to understand changes in daily heavy snowfalls at the scale of European regions and countries. Daily extreme snowfalls result from the interplay of both dynamical and thermodynamic factors, playing at different spatial and time scales: at local (few kms, few hours) scales, geographical features and convection may enhance snowfall precipitations. Persistence of convective snowfalls for several hours on the same region can provide large snowfall amounts detectable at daily time-scales. At synoptic scales, snowfalls are driven by extratropical cyclones ( ~1000 km, 2-6 days) traveling southwards in jet-stream meanders formed by the disruption of the normal westerly flow (Tibaldi et al. 1983, Barnes et al. 2014, Lehmann et al. 2015). Oscillations of the jet stream are associated with low-frequency variability of weather patterns that can modulate daily synoptic fields and snowfall events (Wallace et al. 2006). These conditions create a dipole consisting of high pressure structures over some regions and low pressure systems (extratropical cyclones) travelling southward in other regions."**

Minor editorial comments:

Section 1

Paragraph starting in l40: It would be good to introduces jet variability also in terms of more classical weather regime literature. Suitable references are for instance: Vautard 1990, Michaelangeli et al. 1995, Yiou and Nogaj 2004 , Woollings et al. 2010, Madonna et al. 2017, ....

**Thank you for suggesting this addition, I have added (L44-55): "The most common way to link low frequency variability to weather phenomena is the computation of daily weather regimes (Vautard et al. 1990, Michelangeli et al. 1995).  In Nakamura et al. 1997 and  Yiou et a. 2004, first connections between extreme weather events and weather regimes have been established. Madonna et al. 2017  found a clear link between eddy-driven jet variability and weather regimes in the North Atlantic-European sector. In winter, if blocking high pressure becomes established close to Greenland, cold air from polar latitudes can be advected towards western Europe (North Atlantic Oscillation negative phase (Cattiaux et al. 2010winter). When this weather pattern is associated with extratropical cyclones travelling southward from northern latitudes extreme snowfalls over UK, France, Benelux and the Iberian Peninsula are expected.  If a high pressure ridge (Atlantic Ridge) extends from the Azores Islands towards the Icelandic region or the British isles, cold air coming from Russia or Scandinavia flows in the Mediterranean Sea. This can cause cyclogenesis in the Tyrrhenian  (Genoa lows) or in the Adriatic seas triggering extreme snowfalls over Italy, the Balkans, Greece and Turkey (Buehler et al. 2011)."**

l42: "confined north stream" -> Please check the formulation.

**The sentence has been removed as part of the reorganisation of the introduction**

paragraph starting in l64: Here would be a place to briefly discuss Kawase et al. 2016

**Kawase is now introduced as suggested in the introduction (L70): "For Japan, Kawase et al. (2016) have shown that thermodynamic feedbacks from anthropogenic forcing may enhance extreme snowfalls in future climates via the interaction of the Japan Sea polar air mass convergence zone with the topography. A similar mechanisms exist also for the Mediterranean sea, as recently detailed in D'Errico et al. (2019)" and in the results section (L279)**

Section 2

l85 / 87 / 115: Abbreviations - also common ones - need to be introduced with full words: ERA5, E-OBS, C3S, NUTS, NUST. Please provide a reference where NUTS-2 regions are defined & taken from.

**All acronyms are defined, except for E-OBS that is not an acronym (to the best I could find). For EOBS it has been specified that it is an ECAD dataset. For NUTS regions a reference has been added.**

Section 4

l206, discussion of Figures 7&8, it would help to be more quantitative now and state the actual values. Also Could you compute the country average CAPE & 2mT during the max SF events in both periods and show it as box-and-whisker plots as Figure 6a (or included there)? This would help a lot contrast cases/countries.

**Thank you for the suggestion. The boxplots have been added to Figure6 (panels c) and d) and commented in the text (L212-217)**

discussion of Figures 7&8&9, it would help to have a few more references to the subfigures, as you jump a lot through the panels…

**Where necessary, I have added the precise information about the subpanels**

l221, delete "fully"

**corrected**

l228 you mean Figure 9

**corrected**

l235: Not sure what is meant here, the initially northerly flow on the western flank of the Genoa Low and then cyclonically around back to the Alps in southerly flow? Or the anomalous so southerly flow in 9f? Please check and clarify.

**This part for Switzerland has been removed. However, for the Balkans, at the end of section 4, I now recall the general mechanism for the enhancements of snowfall precipitations in the Balkans (275-280): "The rationale for explaining the changes is then the following: AR patterns happen with the same frequency in winter but associated with deeper cyclogenesis in the Adriatic or Thyrrenian sea. These cyclones find warmer surfaces and availability of humidity and CAPE, thus producing large snowfall amounts, enhanced by stau effect on the Balkans topography. This mechanism is similar to the one described for Japan in Kwase et al (2016) and for Italy in D'Errico et al 2019."**

l272: Clarify that this is only true for the Mediterranean countries, not CH.

**The analysis for Switzerland has been removed, so the text is now coherent.**

Discussion in line 275: The results of Santos et al. 2016 might help to discuss your results more in the light of weather regimes.

**I have added (L308-312): "Although winter total precipitations in future climate scenarios is expected to increase over Europe (Santos et al., 2016), global and regional warming is projected to reduce average and extreme snowfall precipitations at least in Central and Western Europe (de Vries et al., 2014). In the same study, de Vries et al. (2014) find that positive trends in snowfalls could still be observed for mountain areas (Alps and Scandinavia) in warmer climates. This seems coherent with the results found for Japan by Kawase et al. (2016) and in the present study for the Balkans."**

l291: check spelling / citationstyle.

**Spelling has been checked as well as citation style**

**Answers to Reviewer 2**

General comments :

The article has been deeply rewritten, mainly following the Reviewers' comments. In particular the analysis is now performed at country level rather than at regional level, which I definitively support. There are also new analyses on CAPE, 2m temperatures and weather regimes. However I'm still puzzled about several points:

- Although the author claims that the agreement between ERA5 and EOBS is "remarkable" (L. 121), I see in Figures 1-2 quite large differences, and not only for Turkey (the color scale doesn't ease comparison but, e.g., there seems to be around 25-50% difference in all Scandinavia). It seems to me that ERA5 tends to overestimate EOBS.

**Following the comment of the reviewer, the sentences have been rephrased (L128-130) as: "The agreement between the ERA5 and the E-OBSv20.0e data-set largely depends on the regions considered. Overall, the climatologies of snowfall provided by the two datasets have similar ranges although ERA5 tends to overestimate EOBSv20.0e"**

- All the results of section 4 to explain positive trends in extreme snowfall are based on four countries : Albania (AL), Macedonia (ME), Switzerland (CH) and Turkey (TR). These countries are chosen because they show "the largest positive changes" in maximum snowfall in ERA 5 (L. 173). However according to Figure 4, three of these countries (namely CH, ME, TR) show strong negative trends in EOBS ! Therefore how much confidence can we have in the results for explaining positive trends? Note that the positive trend in ERA5 for Switzerland is in contradiction with several other studies based on snow stations (e.g. Scherrer et al. 2013, Marty and Blanchet, 2012)

**This comment is common for Reviewer 1,2 and the editorial boarding. In the new version of the manuscript I therefore focus only on the countries showing consistent positive trends in EOBS and ERA5, namely Albania, Bosnia and Montenegro. For Switzerland, as pointed out by the referee, it has been added that: " the positive trend in ERA5 for Switzerland is in contradiction with several other studies based on snow stations (e.g. Scherrer et al. 2013, Marty and Blanchet, 2012)"**

- What the author calls "Macedonia" in all the article seems actually to be the Montenegro (which indeed shows a positive trend in both ERA5 and EOBS)

**Thank you for noticing this error. It is indeed Montenegro. This has been fixed through the manuscript.**

- According to Figure 4, there are only 3 countries with strong positive trends in both ERA5 and EOBS: Bosnia, Montenegro, Albania. All three are located on the eastern flank of the Adriatic Sea. Therefore I suggest focusing on these three countries to attempt explaining positive trends.

**Following the suggestion of the reviewer, all the analysis now focused on these three countries**

- I didn't understand the results of the weather regime analysis. Whereas AL and ME show similar sea level pressure fields for the maxima, CH and TR show very different fields. Therefore isn't it inconsistent that for the four countries AR and BLO are found to be by far the most frequent weather regimes during the days of maximum snowfalls?

**The analysis is now focused on AL,BO,ME so this problem is fixed. However, I would like to stress that although with some differences in the position of the cyclonic patterns over Europe, snowfalls in CH and TR are also associated with blocking or AR weather regimes, as described in the introduction. Indeed NAO+ or NAO- are zonal patterns characterized by westerly (NAO+) or easterly (NAO-) associated respectively with wet-mild (NAO+) conditions or cold-dry (NAO-) conditions.**

Detailed comments:
- L 121 "is remarkable": as said above it seems to me there are quite large differences for different countries. To ease comparison I suggest 1) using a color scale with larger gradient, 2) showing the ratio ERA5/EOBS.

**Thank you for the suggestion. I have removed remarkable and rephrased the sentence. I have also added in Figure 5, two panels showing the symmetric percentage error (x_ERA-x_EOBS)/(x_ERA+x_EOBS), where x is the average snowfall. This quantity is normalized in the range [-1, 1] and it shows that for southern Europe ERA is positively biased and for Central Europe is negatively biased (see discussion in L162-170).**

- Figure 1c: why is part of Turkey missing ?

**Thank you for noticing it. Part of the figure was missing because of some NaN present in the datasets. The average is now obtained by removing the NaN and the figures are fixed. This causes a change in the averaged values of the climatology that were underestimated. The trends however did not change in any appreciable way except for the magnitude.**

- L 162 "notably ion the Balkans": I'd be more moderated on the Balkans because EOBS and ERA5 only agree on 3 countries.

**"Notably in the Balkans" has been replaced with "specifically for some countries in the Balkans".**

- L 173: "the largest positive changes, namely Albania (AL), Macedonia (ME), Switzerland (CH) and Turkey (TR)": as already said, among these countries actually ERA and EOBS only agree on AL…

**As previously said, the analysis now focuses on AL, ME and BO**

- In all Section 4, don't you mean Montenegro rather than Macedonia?

**Thank you for noticing this error, that has now been corrected**

- Figure 6 and L. 181-184: the difference are so small that it seems to me to be only sampling variability.

**I would tend to agree, but the subsequent analyses show that the differences are associated with precise sea-level pressure patterns, temperature and CAPE anomalies**

- L 188 "for Switzerland maxima of snowfalls tend to occur in December rather than in January": Yes, this is in accordance with Klein et al. 2016.

**Thank you, but the analysis for Switzerland has been removed**

- Figures 7-8-9: the green dots hide where we are supposed to look at. Please consider replacing the dots by the country borders.

**Thank you for the suggestion. After several attempts, I still prefer this choice for presenting the figures. The new boxplots in Figure 6c,d, suggested by the editor, provide a quantitative "zoom" on the countries selected**

- L 207 "and on the Alps for Switzerland": actually for CH, CAPE is large not only over the Alps, but rather over all eastern Europe.

**The analysis for Switzerland has been removed**

- L 214-215 "the local temperature difference … is small" : what would be large? For AL the difference is of 3°C. Isn't it large? For CH: is it red or white? The green dot is about the size of the country so it doesn't allow us to see the values.

**Thank you for the remark, the analysis of the boxplots in Figure 6c,d allows now for more quantitative statements (L212-217): "The boxplots in Figure 6c-d) show the spatial regional average of t2m and CAPE during the days of the maximum snowfalls. The analysis for temperature (Figure 6c) suggests that maximum snowfalls tend to be associated with temperatures above the freezing point in the recent period and a reduced variability with respect to the first period. The analysis for CAPE (Figure 6d) is not very informative for two reasons: i) the distribution is highly non-Gaussian, it includes zeros and presents several outliers, ii) convective precipitations can originate in nearby regions and be transported. In**

- L 235 "increase of CAPE over the central Med … Switzerland": I don't understand this interpretation. According to Figure 9, the maxima in CH are produced by Atlantic flows, so I don't get the interpretation in terms of Mediterranean CAPE.

**The analysis for Switzerland has been removed**

- L 245 "technique presented in Faranda et al. 2017": do you take the same thresholds as therein to produce the classification (98% and 2%)?

**Yes, this is now specified in the text**

- L 245 "five possible regimes" It would be helpful to show the composites of the five classes.

**The composite are shown in Faranda et al. 2017 for NCEP but they are very similar for ERA5. Since the paper already counts several multipanel figures, the regimes are not shown but a precise reference to the figure 2 in Faranda et al 2017 is made.**

- Figure 10 and L 248 "prevalence of BLO and AR patterns": as already said, I don't understand how BLO and AR can prevail for CH and TR whereas Figure 9 shows completely different influences for these countries.

**Thank you, but the analysis for Switzerland has been removed**

- L 251-254 "this patterns consists … Mediterranean basin": according to Figure 9, this doesn't seem to apply to CH.

**Thank you, but the analysis for Switzerland has been removed**

Typos and clarifications:
- L 147: positive IN Switzerland
- L 185 Figure 6 → Figure 6b
- L 186 date → average date?
- L 207 SwitZerland

**Wherever still presents, these points have been fixed.**

[revised manuscript text omitted]

---

## Author Response (AR3)

Dear Editor,

I have taken into account the final comments and I am happy to provide a new version of the manuscript, hopefully ready for publication.  Thank you again for checking the manuscript consistency and suggesting the below mentioned corrections.

Kind Regards,
Davide Faranda

Before submitting the final version please carefully check for spelling / typping / citation format errors e.g. in lines 45, 58, 276 (their -> they?)

**These and other typos have been fixed.**

The shading in Figure 3 is literally not visible

**The shading of Figures 3 and 4 has been changed to green to improve visibility.**

The summary in lines 170-177 really reads nice!

**Thank you!**

l209: it is remarkable to see snowfally at 10°C perhaps you should comment on this.

**I have added: "It is not unusual to observe snowfall with ambient temperatures up to 6∘C (see, e.g. Steinacker (1983)) and more recent studies show that snow and rain can coexist even up to 13.3∘C (Wen215et al. 2013)." LL  214-216**

l215: The statement about CAPE is quite negative here, although it later turns out to be useful to use. Rethink if the figure in that way is needed, if you later have to conclude that CAPE is rather relevant upstream of the snowfall region.

**It is true that the statement about CAPE is negative here, however I think it is important to stress the difference between local CAPE and the discussion which follows about transported convection. I have added "local" to the sentence  in L216-218.**

Figure 8 l227: The region you are referring to is hard to see. perhaps zoom the plot on the Adriatic region?

**Here I have added that I was referring to the results of Figure 6c) besides the maps in Figure 8. I am a bit uncomfortable in having Figures (7,8,9) with different levels of zoom. The boxplots in Figure 6 should support the readers in "zooming" the results on the (rather small) countries considered.**

l277: this is now nicely elaborated and might be worth to repeat in the conclusions.

**I have added "in presence of sea-to-land flow associated with extratropical cyclones" in LL307**

There was one remaining comment by a reviewer : thank you again for your much effort to revise the manuscript. I have no additional comments before acceptance except to remove Nakamura et al. (1997) from L45.

**This reference has been now removed**

[revised manuscript text omitted]